



# Factors controlling the community structure of picoplankton in contrasting marine environments

Jose Luis Otero-Ferrer[1], Pedro Cermeño[2], Bieito Fernández-Castro[1,3], Josep M. Gasol[2,5], Xosé Anxelu G. Morán[4], Emilio Marañon[1], Víctor Moreira-Coello[1], Marta Varela[6], Marina Villamaña[1], and Beatriz Mouriño-Carballido[1]

[1]Departamento de Ecoloxía e Bioloxía Animal, Universidade de Vigo, Spain
[2]Institut de Ciències del Mar, Consejo Superior de Investigaciones Científicas-Barcelona, Spain
[3]Departamento de Oceanografía, Instituto de investigacións Mariñas (IIM-CSIC), Vigo, Spain
[4]King Abdullah University of Science and Technology (KAUST), Read Sea Research Center, Biological and Environmental Sciences and Engineering Division, Thuwal, Saudi Arabia.
[5]Centre for Marine Ecosystem Research, School of Sciences, Edith Cowan University, WA, Australia
[6]Centro Oceanográfico de A Coruña, Instituto Español de Oceanografía, A Coruña, Spain.

*Correspondence to:* Jose Luis Otero-Ferrer (jootero@uvigo.es)

**Abstract.** The effect of inorganic nutrients on planktonic assemblages has been traditionally relied on concentrations rather than estimates of nutrient supply. We combined a novel dataset of hydrographic properties, turbulent mixing, nutrient concentration and picoplankton community composition with the aim of: i) quantifying the role of temperature, light and nitrate fluxes as factors controlling the distribution of autotrophic and heterotrophic picoplankton subgroups, as determined by flow cytometry; and ii) describing the ecological niches of the various components of the picoplankton community. Data were collected in 97 stations in the Atlantic Ocean, including tropical and subtropical open ocean waters, the northwestern Mediterranean Sea, and the Galician coastal upwelling system of the Northwest Iberian Peninsula . A Generalized Additive Model (GAM) approach was used to predict depth-integrated biomass of each picoplankton subgroup based on three niche predictors: sea surface temperature, averaged daily surface irradiance, and the transport of nitrate into the euphotic zone, through both diffusion and advection. In addition, niche overlap between different picoplankton subgroups was computed using non-parametric kernel density functions. Temperature and nitrate supply were more relevant than light in predicting the biomass of most picoplankton subgroups, except for *Prochlorococcus* and low nucleic acid (LNA) prokaryotes, for which irradiance also played a significant role. Nitrate supply was the only factor that allowed the distinction between the ecological niches of all autotrophic and heterotrophic picoplankton subgroups. *Prochlorococcus* and LNA prokaryotes were more abundant in warmer waters (> 20ºC) where the nitrate fluxes were low, whereas *Synechococcus* and high nucleic acid (HNA) prokaryotes prevailed mainly in cooler environments characterized by intermediate or high levels of nitrate supply. Finally, the niche of picoeukaryotes was defined by lower temperatures and high nitrate supply. We estimated that with the predicted decrease in nitrate supply as the result of global change, a 9% increase in the ratio cyanobacteria to picoeukaryotes would occur in a future ocean, something that would have important implications, given the contribution of smallest cells to the biological carbon pump. These results support the key role of nitrate supply, as it not only promotes the growth of large phytoplankton, but it also controls the structure of marine picoplankton communities.





# 1 Introduction

Picoplankton, including archaea, bacteria and picoeukaryotes are the smallest (cell diameter <2 μm) and most abundant organisms in marine ecosystems. Photosynthetic picoplankton often dominate marine phytoplankton biomass and primary production in oligotrophic tropical and subtropical regions (Chisholm, 1992), whereas they are typically a minor component in nutrient-replete coastal environments, usually dominated by large-sized plankton species (Finkel et al., 2010; Marañón, 2015). However, due to the large temporal and spatial variability in the structure and composition of the microbial community in shelf seas (Sherr et al., 2005), picoplankton, together with nanoplankton, can dominate the microbial community under certain conditions (Morán, 2007; Espinoza-González et al., 2012). In addition, picoplankton contributes overwhelmingly to the recycling of organic matter (Azam et al., 1983; Fenchel, 2008), and previous studies suggest that photosynthetic picoplankton could also play a role in the export of carbon to the deep ocean (Richardson and Jackson, 2007; Lomas and Moran, 2011; Guidi et al., 2015). As a result, picoplankton is considered a key component of the current carbon cycle and likely more important in future climate warming scenarios (Laufkötter et al., 2016).

When analyzed by flow cytometric techniques, two genera of picocyanobacteria (*Prochlorococcus* and *Synechococcus*), one or two subgroups of autotrophic picoeukaryotes of different size (small and large), and two subgroups of heterotrophic prokaryotes, based on their high (HNA) or low nucleic acid (LNA) content, can be distinguished (Gasol and del Giorgio, 2000; Marie and Partensky, 2006). Although closely related phylogenetically, *Synechococcus* and *Prochlorococcus* exhibit distinct physiological traits (Moore et al., 1995), divergent evolutionary strategies (Scanlan and West, 2002) and disparate geographic distributions (Zubkov et al., 2000). *Prochlorococcus* tend to be restricted to relatively warm (above 15ºC) and nutrient-poor waters, extending from the surface down to 150 m, along the 40º N–40º S latitudinal band (Partensky et al., 1999b; Johnson et al., 2006). *Synechococcus* exhibit a wider geographic and thermal distribution, including high-nutrient waters and occasionally reaching polar latitudes (Paulsen et al., 2016), their vertical distribution being shallower than that of *Prochlorococcus* (Partensky et al., 1999a; Li, 2002). The contribution of picoeukaryotes to picoplankton biomass is generally smaller than the contribution of picocyanobacteria (Zubkov et al., 2000; Buitenhuis et al., 2012), except in coastal regions where their contribution usually increases (Grob et al., 2007). In general, LNA prokaryotes dominate heterotrophic prokaryotic biomass in the oligotrophic open ocean, whereas HNA cells dominate in coastal regions (Li et al., 1995; Bouvier et al., 2007). These contrasting spatial distributions suggest that the picoplankton subgroups occupy differential ecological niches or, according to the classical definition proposed by Hutchinson (1957), distinct multidimensional hypervolumes of environmental factors in which viable populations develop. By describing the overlaps of environmental factors, realized niche partitioning can be defined, and the factors controlling the distribution of picoplankton subgroups can be identified. However, despite decades of experimental and field observations, the relative importance of the factors driving the variability of the growth and the spatial distribution of picoplankton subgroups remain largely unknown.

Aside from the effect of trophic controls, the distribution of microbial plankton is primarily determined by seawater tempera-





ture, light and nutrients (Li, 2009; López-Urrutia, 2013; Barton et al., 2015). Quantifying their relative influence on the spatial and temporal distribution of the different picoplankton subgroups is complicated by the fact that the above mentioned factors are often correlated in the ocean (Finkel et al., 2010). This shortcoming can be circumvented by using experimental approaches in the laboratory, where the influence of each independent factor is isolated. Alternatively, it can be approached by combining

large data sets of hydrographic and biological observations collected from contrasting marine environments, which allow us to characterize the suite of variables that best define the organism's ecological niches.

In order to study the significance of temperature and nutrient concentrations in determining the contribution of picophytoplankton to total phytoplankton biomass and production, Agawin et al. (2000) reviewed the available literature from oceanic and coastal estuarine areas. Although the number of observations for which both temperature and nutrient concentration were

available was too small to statistically separate their effects, these authors hypothesized that the dominance of picoplankton in warm, oligotrophic waters was due to differences between picophytoplankton and larger cells in the capacity to use nutrients and in their intrinsic growth rate. Bouman et al. (2011) investigated how vertical stratification controls the community structure of picophytoplankton in subtropical regions. According to their results, photosynthetic picoeukaryotes dominate in weakly-stratified, whereas in strongly stratified waters, *Prochlorococcus* cyanobacteria are prevalent. More recently, Flombaum et al.

(2013), using a compilation of flow cytometry data from all major ocean regions, concluded that *Prochlorococcus* and *Synechococcus* abundance distributions were controlled by temperature and photosynthetically active radiation (PAR, 400-700 nm), discarding the role of nitrate concentration.

However, in tropical and subtropical domains, the best represented regions in Flombaum et al.'s study, surface nitrate is almost depleted and the variability of its concentration can be widely disconnected from changes in its availability for phytoplankton,

which depends more on the supply from deeper waters by turbulent diffusion (Mouriño-Carballido et al., 2016). It is also believed that fine-scale turbulence can enhance the nutrient uptake and subsequent growth of larger phytoplankton (Lazier and Mann, 1989; Karp-Boss et al., 1996; Guasto et al., 2012), especially in regions with low nutrient levels and strong grazing pressure (Barton et al., 2014).

As far as we know, only one study has previously used estimates of nitrate availability, derived from observations of mi-

crostructure turbulence, to investigate the role of nutrient availability in controlling the composition of picoplankton communities(Mouriño-Carballido et al., 2016). These authors, using local data from the northwestern Mediterranean Sea, found that different autotrophic picophytoplankton subgroups exhibit contrasting responses to nitrate supply, and that as a result the ratio of prokaryotic to picoeukaryotic photoautotrophic biomass decreased with increasing nitrate supply. However, whether these patterns are general and widespread in the ocean remains largely uncertain, given that no concomitant datasets including mea-

surements of turbulent diffusion nutrient flux and picoplankton subgroup structure have been available to date.

Here we extend the analysis described in Mouriño-Carballido et al. (2016) by combining a dataset of picoplankton community composition, hydrographic properties, turbulent mixing and inorganic nutrient concentrations collected in a total of 97 stations. Observations were made in contrasting environments of the Atlantic Ocean, in order to quantify the role of temperature, light and nitrate availability in the composition of the picoplankton community, and to describe the ecological niches of each pi-




coplankton subgroup.

## 2 Material and Methods

This study includes data collected in 97 stations from three contrasting environments covering the tropical and subtropical
Atlantic Ocean (T), the northwestern (NW) Mediterranean Sea (M), and the Galician coastal upwelling ecosystem (G), be-
tween October 2006 and December 2015 (see Table 1 and Figure 1). Two cruises (CARPOS Oct-Nov 2006 and TRYNITROP
Apr-May 2008) sampled 26 stations located in the tropical and subtropical Atlantic Ocean. Three other cruises carried out in
the NW Mediterranean Sea (FAMOSO1 Mar 2009, FAMOSO2 Apr-May 2009, and FAMOSO3 Sep 2009) sampled 19 stations
during three contrasting hydrographic conditions, covering from winter mixing to summer stratification. Finally, 52 stations
were sampled in the Galician coastal upwelling ecosystem during HERCULES1 Jul 2010, HERCULES2 Sep 2011, HER-
CULES3 Jul 2012, DISTRAL Feb-Nov 2012, ASIMUTH Jun 2013, CHAOS Aug 2013, and NICANOR Feb 2014-Dec 2015
cruises. Additional information about the sampling design of these cruises is presented in Aranguren-Gassis et al. (2011, CAR-
POS), Mouriño-Carballido et al. (2011, TRYNITROP), Mouriño-Carballido et al. (2016, FAMOSO), Cermeño et al. (2016,
DISTRAL), Villamaña et al. (2017, CHAOS), Díaz et al. (under revision 2018, ASIMUTH) and Moreira-Coello et al. (2017,
NICANOR).

At each station, information about hydrographic properties, turbulent mixing, nitrate concentration and picoplankton commu-
nity composition were collected. Light conditions for each sampling station were considered as the 5-day averaged daily surface
photosynthetic active radiation (PAR) obtained from satellite data (http://globcolour.info). Light attenuation coefficients were
obtained from vertical profiles of estimated by Licor sensors using the Beer-Lambert Law equation (Kirk, 1994). Depth of
the euphotic layer was calculated as the depth where PAR was 1% of its surface value. For those cruises where PAR profiles
were not available (ASIMUTH, CHAOS and NICANOR), the depth of the photic layer was calculated by considering light
attenuation coefficients derived from surface chlorophyll-*a* data estimated from the space, following the algorithms proposed
by Morel et al. (2007) (http://globcolour.info).

### 2.1 Hydrography and turbulent mixing

Hydrographic properties and turbulent mixing were derived from a microstructure turbulent profiler (Prandke and Stips, 1998,
MSS) equipped with a high-precision Conductivity-Temperature-Depth (CTD) probe, two microstructure shear sensors (type
PNS06), and also a sensor to measure the horizontal acceleration of the profiler. Measurements of dissipation rates of turbulent
kinetic energy ($\varepsilon$) were conducted at 3-10 profiles for each station to the bottom, or 300 m over deep waters. The profiler was
balanced to have negative buoyancy and a sinking velocity of 0.4 to 0.7 m s$^{-1}$. The frequency of data sampling was 1024 Hz.
The sensitivity of the shear sensors was checked after each use. Due to significant turbulence generation close to the ship, only
the data below 5 (HERCULES1, HERCULES2, HERCULES3, DISTRAL, ASIMUTH, CHAOS, and NICANOR) and 10 m





(CARPOS, TRYNITROP, FAMOSO1, FAMOSO2, FAMOSO3) were considered reliable. Data processing and calculation of dissipation rates of ($\varepsilon$) was carried out with the commercial software MSSpro. The squared Brunt Väisälä frequency ($N^2$) was computed from the CTD profiles according to the equation:

$$N^2 = -\left(\frac{g}{\rho_w}\right)\left(\frac{\partial\rho}{\partial z}\right)(s^{-2}) \tag{1}$$

where g is the acceleration due to gravity (9.8 m s$^{-2}$), $\rho_w$ is seawater density (1025 kg m$^{-3}$), and $\partial\rho/\partial z$ is the vertical potential density gradient. Vertical diffusivity ($K_z$) was estimated as:

$$K_z = \Gamma\frac{\varepsilon}{N^2}(m^2 s^{-1}) \tag{2}$$

where $\Gamma$ is the mixing efficiency, here considered as 0.2 (Osborn, 1980).

## 2.2   Nutrient supply

Samples for the determination of nitrate ($NO_3$) + nitrite ($NO_2$) were collected from 3 to 9 depths in rinsed polyethylene tubes and stored frozen at -20 °C until analysis on land, according to standard methods using the automated colorimetric technique (Grasshoff et al., 2007). In those stations carried out during the NICANOR cruises, where nitrate concentrations were not available, they were obtained by using nitrate-density relationships (Moreira-Coello et al., 2017). Vertical diffusive fluxes of

nitrate into the euphotic zone were calculated following the Fick's law as:

$$FluxNO_3 = \bar{K}_z\Delta NO_3 \tag{3}$$

where $\Delta NO_3$ is the nitrate vertical gradient obtained by linear fitting of nitrate concentrations in the nitracline, determined as a region of approximately maximum and constant gradient, and ($\bar{K}_z$) is the averaged turbulent mixing over the same depth interval. In the Galician coastal upwelling, nitrate diffusive fluxes were estimated over a fixed depth interval using the same

procedure (20-40 m) except in the shallowest stations where we compute the surface-bottom flux.

Most stations carried out in the Galician coastal upwelling were conducted inside three different Rías (Ría de Vigo, Ría de Pontevedra and Ría de A Coruña). The Rías are coastal embayments affected by seasonal wind-driven coastal upwelling of cold, nutrient-rich North Atlantic Central water (Wooster et al., 1976; Fraga, 1981; Álvarez-Salgado et al., 1993). The total nitrate supply in the Galician Rías was computed as the sum of nitrate vertical diffusion plus nitrate vertical advection

due to coastal upwelling. A simplified estimate of nitrate supply through vertical advection due to upwelling was computed considering the Galician Rías as single boxes divided into two layers (Álvarez-Salgado et al., 1993), the deeper one influenced by upwelled inflowing waters and the surface layer dominated by the outgoing flow. Assuming that the bottom layer volume is conservative and stationary, the vertical advective flux ($Q_Z, m^3 s^{-1}$), would be equivalent to the incoming bottom flux ($Q_B, m^3 s^{-1}$), computed as the product of the upwelling index ($I_W, m^3 s^{-1} km^{-1}$) and the lengths of the mouth of the Rías

(ca. 10-11.5 km). $I_W$ was averaged over the three-day period before each cruise from wind data recorded by meteorological buoys located in Cabo Vilano (HERCULES, NICANOR) and Cabo Silleiro (DISTRAL, ASIMUTH, CHAOS, ASIMUTH), or



modeled by the Fleet Numerical Meteorology and Oceanography Center (FNMOC) model when buoy data were not available (http://www.indicedeafloramiento.ieo.es). Finally, the transport of nitrate into the euphotic zone through vertical advection was computed as:

$$NO_3\,Advective\,flux = \frac{Q_z}{A_{basin}}[NO_3]_D \tag{4}$$

where $A_{basin}$ is the surface area of the Galician Rías, $Q_Z$ is the vertical advective flux, and $[NO_3]_D$ is the averaged nitrate concentration at the base of the euphotic layer. $A_{basin}$ is 141 km$^2$ for Ría de Pontevedra (ASIMUTH), 174 km$^2$ for Ría de Vigo (CHAOS, ASIMUTH, DISTRAL), and 145 km$^2$ for Ría de A Coruña (HERCULES, NICANOR) (see Villamaña et al. 2017 and Moreira-Coello et al. 2017, for details).

## 2.3 Flow cytometry

Picoplankton samples (1.8 ml) for the determination of picoplankton abundance and cell properties were taken from 3-9 depths and measured immediately on board (TRYNITROP), or preserved with paraformaldehyde plus glutaraldehyde (P+G) and frozen at $-80°$ C until analysis in the laboratory (the other cruises). Unfortunately, due to problems with sample preservation, only heterotrophic or autotrophic picoplankton subgroup data were available for the DISTRAL and ASIMUTH cruises,

respectively. Two aliquots from the same sample were used for the study of picophytoplankton (0.6 ml) and heterotrophic prokaryotes (0.4 ml), analyzed at high (mean 60 $\mu$l min$^{-1}$) and low (mean 18 $\mu$l min$^{-1}$) flow rates during 4 and 2 min, respectively. Before the analysis, the DNA of heterotrophic prokaryotes was stained with nucleic acid-specific fluorescent dye (SYTO-13 or SYBR1). A FACSCalibur flow cytometer (Becton-Dickinson) equipped with a laser emitting at 488 nm was used to measure and count picoplankton. Autotrophic cells were separated into two groups of cyanobacteria (*Synechococcus*

and *Prochlorococcus*) and one group of small picoeukaryotes, based on their fluorescence and light scatter signals (SSC), as explained in Calvo-Díaz et al. (2006). Two groups of heterotrophic prokaryotes (LNA and HNA) were distinguished based on their relative green fluorescence, which was used as a proxy for nucleic acid content (Gasol and del Giorgio, 2000; Bouvier et al., 2007).

## 2.4 Generalized Additive Models

A Generalized Additive Model (GAM) approach was used to predict depth-integrated biomass of each picoplankton subgroup, the contribution of LNA prokaryotes to heterotrophic picoplankton, the cyanobacteria to picoeukaryotes ratio, and the autotrophic to heterotrophic ratio based on observations and estimates of three environmental factors: sea surface temperature (SST), daily surface PAR, and the transport of nitrate into the euphotic zone (NO$_3$ Flux), including both diffusive and advective

processes. GAMs assume that the effect of each predictor on the response variable can be described by smoothed functions whose effects are additive. Due to the large number of zero observations, data overdispersion, and the need for a single parsimonious model to make predictions for a large number of groups, we assumed that the depth-integrated biomass of each



picoplankton subgroup, relative contribution values and biomass ratios all followed negative binomial distributions. Those niche descriptors that did not follow normal distributions were log transformed. The complete model structure for the biomass of each picoplankton subgroup was:

$$y_j = I + s(SST) + s(PAR) + s(\log(NO_3 Flux)) + Error \tag{5}$$

where $y$ represents the depth-integrated biomass for each picoplankton subgroup $j$, and s a cubic regression spline used for fitting the observations to the model (Wood, 2006).

Generalized models include a function linking the mean value of $y_j$ and the predictors. For those response variables that followed a negative binomial distribution the used link function was the natural logarithm. The LNA contribution to total heterotrophic prokaryotes was adjusted using a gaussian distribution and an identify link (Wood et al., 2016). The inclusion of

10 the different predictors to explain the response variable (the biomass of each picoplankton subgroup, its relative contribution and biomass ratios) was assessed via stepwise model selection using the minimum Akaike Information Criterion (Hastie and Tibshirani, 1993).

Smooth terms were tested using a Bayesian test (Marra and Wood, 2012) to prevent overfitting. GAMs were evaluated based on explanatory power (explained variance) and goodness-of-fit (GOF). GOF was assessed via quantile–quantile (QQ) plots of

15 Pearson residuals (provided in SM Figure A1). All calculations were done using the mgcv package (Wood, 2011) in R (R Core Team, 2015).

## 2.5 Niche overlap analysis

The estimation of niche overlap between different picoplankton subgroups based on non-parametric kernel density functions

($NO_K$) was calculated following Mouillot et al. (2005):

$$NO_{K\,i,j,t} = 1 - \frac{1}{2} \int |f_{it}(x) - f_{jt}(x)| \, dx \tag{6}$$

where $NO_{K_{i,\,j,\,t}}$ is the niche overlap between picoplankton subgroups $i$ and $j$ for the environmental factor $t$, and $f_{it}$ and $f_{jt}$ are the kernel population density functions of factor $t$ for species $i$ and $j$, respectively. In order to correct the correlation between niche predictors, we used the estimator in a dependent sample (EDS) proposed by Kark et al. (2002).

To assess the statistical niche differences between subgroups, null model permutation tests were performed to verify whether the niche overlaps were significantly lower than 100% (Geange et al., 2011). When the contribution of depth-integrated biomass for each picoplankton subgroup exceeded that expected by chance (1/3 for autotrophic and 1/2 for heterotrophic picoplankton), niche predictors for each station were selected. Statistical null distributions (the distribution of the statistic test under the null hypothesis of no niche differentiation) were generated by calculating pseudo-values through randomly permuting group

labels in the corresponding data set over 10000 runs. The distributions of the average niche overlaps for the null model were then computed. Niche overlap calculations and associated null model tests were performed using the density function and the source code provided as supporting information in Geange et al. (2011). All calculations were done using R (R Core Team,



2015).

## 2.6 Present and future spatial distributions of the cyanobacteria to picoeukaryote biomass ratio

A rough estimation of the spatial distribution of the cyanobacteria to picoeukaryotic depth-integrated biomass ratio was calculated by first estimating a proxy for nitrate diffusive flux based on SST, the vertical gradient of nitrate across the nitracline and surface chlorophyll-*a* applying a GLM model (Adj-$R^2$=0.31, p< 0.01, n = 147). This relationship was determined with data from all stations included in our dataset plus 50 additional stations visited during the Malaspina circumnavigation expedition (Fernández-Castro et al., 2015). We then used this relationship to compute the present large-scale spatial distribution of nitrate diffusive supply based on sea surface temperature and chlorophyll-*a* provided by NASA (National Aeronautics and Space Administration, http://oceandata.sci.gsfc.nasa.gov/MODIS-Aqua/Mapped/Annual/9km/) and nitrate gradient derived from the World Ocean Atlas (WOA09) provided by NOAA (National Oceanographic and Atmospheric Administration). Finally, these estimates of nitrate diffusive supply were used as input variables in the GAM model derived from our study to compute the spatial distribution of the cyanobacteria to picoeukaryotic biomass ratio for the present scenario. Only nitrate supply was used as a predictor, as this factor explained 51% (Adj-$R^2$=0.35, p< 0.001, n = 97) of the variability. To compute the same ratio in a future global change scenario (year 2100), we followed the same procedure but used the averaged global decrease in dissolved nitrogen flux into the euphotic zone estimated by Lewandowska et al. (2014) between years 2000 and 2100 by using global ocean model simulations (ca. 20%). The percentage change in the cyanobacteria to picoeukaryote biomass ratio was calculated based on the difference between the annual averages of future and present estimates.

## 3 Results

### 3.1 Environmental variables and picoplankton biomass

Our database covered a wide environmental gradient from oligotrophic to eutrophic conditions. Stations sampled in the tropical and subtropical Atlantic oceans (T) were, on average, characterized by warm surface waters (26 $\pm$ 2 ℃, mean $\pm$ SD) where the supply of nitrate through vertical diffusion from deeper waters was low (0.7 $\pm$ 1.6 mmol N m$^{-2}$ d$^{-1}$), and surface chlorophyll-*a* was low (0.1 $\pm$ 0.1 mg m$^{-3}$) (Table 2 and Figure 2). The NW Mediterranean Sea (M), sampled from March to September, was characterized by cooler surface waters (16 $\pm$ 4 ℃), and intermediate values of both nitrate vertical diffusive supply (41 $\pm$ 113 mmolN m$^{-2}$ d$^{-1}$) and also surface chlorophyll-*a* (0.9 $\pm$ 0.9 mg m$^{-3}$). Finally, the stations sampled in the Galician coastal upwelling system (G), which included year-round samples, were characterized by relatively cold surface waters (16 $\pm$ 2 ℃), enhanced rates of nitrate supply (30 $\pm$ 46 mmolN m$^{-2}$ d$^{-1}$) and high values of surface chlorophyll-*a* (2.2 $\pm$ 2.5 mg m$^{-3}$). No statistically significant differences were observed in averaged surface photosynthetic active radiation between the three regions.



Differences in picoplankton biomass and composition were also observed between the three domains. Averaged photic layer depth-integrated total picoplankton biomass (including both LNA and HNA prokaryotes, *Prochlorococcus*, *Synechococcus* and picoeukaryotes) was higher in the tropical and subtropical Atlantic ($1052 \pm 215$ mg C m$^{-2}$) and NW Mediterranean ($1038 \pm 485$ mg C m$^{-2}$), compared to the Galician coastal upwelling ($216 \pm 36$ mg C m$^{-2}$) (Table 2). In the tropical and subtropi-

5  cal Atlantic, the contribution of *Prochlorococcus* to total picoplankton biomass was 41%, followed by LNA (27%) and HNA (22%) prokaryotes, with smaller contributions of *Synechococcus* and picoeukaryotes (<5% each). In the NW Mediterranean, picoplankton biomass was on average dominated by *Synechococcus* (50%), followed by LNA and HNA prokaryotes (∼17% for each group), picoeukaryotes (10%) and *Prochlorococcus* (5%). Finally, HNA (55%) and LNA (21%) prokaryotes dominated in the Galician coastal upwelling system, followed by picoeukaryotes (11%), *Synechococcus* (6%) and *Prochlorococcus* (1%).

## 3.2   The role of environmental factors in picoplankton composition

In order to explore the role of temperature, light and nitrate supply in the composition of the picoplankton community, we first used Generalized Linear Models (GLM) to investigate simple linear relationships between each of these factors and the depth-integrated biomass of each picoplankton subgroup, the contribution of LNA prokaryotes to heterotrophic picoplankton biomass, the cyanobacteria (*Prochlorococcus* and *Synechococcus*) to picoeukaryotes ratio, and the autotrophic to heterotrophic

picoplankton ratio (Figure 3 and Table 3). All picoplankton groups except picoeukaryotes exhibited statistically significant relationships with sea surface temperature. This relationship was negative for *Synechococcus*, and positive for all the other subgroups (Figure 3). Only LNA and HNA prokaryotes and *Synechococcus* exhibited significant, positive relationships with surface radiation. All groups except the picoeukaryotes were negatively correlated with nitrate fluxes. The contribution of

LNA prokaryotes to heterotrophic picoplankton biomass only exhibited a significant negative relationship with nitrate fluxes, whereas the cyanobacteria to picoeukaryotes ratio was positively correlated with surface temperature and negatively with nitrate fluxes. Finally, the ratio of autotrophic to heterotrophic biomass was not linearly correlated with any of the studied environmental factors.

In order to exclude cross-correlation between the three environmental factors and consider the possibility of non-linear rela-

tionships, we subsequently fitted the data to Generalized Additive Models (Figure 4 and Table 3). Temperature was the only factor included in the models built for predicting the depth-integrated biomass of all picoplankton subgroups. HNA prokaryotes exhibited a positive relationship with temperature above 19ºC, whereas *Prochlorococcus* and LNA prokaryotes showed a nearly sigmoid curve relationship with a transition between ca. 16 ºC and 25 ºC (Figure 4). The relationship between the biomass of both *Synechococcus* and picoeukaryotes and temperature showed a negative trend until ∼20ºC, and remained relatively con-

stant above this temperature. PAR was included in the models of all picoplankton groups except picoeukaryotes. Whereas the biomass of *Prochlorococcus* exhibited a saturation type relationship with PAR, heterotrophic prokaryotes and *Synechococcus* showed a linear positive relationship. Finally, only LNA prokaryotes*, Synechococcus* and picoeukaryotes exhibited statistically significant relationships with nitrate supply. This relationship was negative for LNA prokaryotes and *Synechococcus*, whereas picoeukaryotes showed a unimodal function, peaking at ∼1 mmol NO$_3$ m$^{-2}$ d$^{-1}$. Nitrate flux was the only factor selected in



the models to predict the contribution of LNA prokaryotes to heterotrophic picoplankton biomass, and both the cyanobacteria to picoeukaryotes biomass ratio and the autotrophic to heterotrophic biomass ratio. This relationship was negative in the three models. Temperature was also negatively correlated with the contribution of LNA prokaryotes to heterotrophic biomass, and the ratio of autotrophic to heterotrophic biomass. Finally, PAR showed a positive correlation with the contribution of LNA

prokaryotes to heterotrophic biomass, and the cyanobacteria to picoeukaryotes biomass ratio.

### 3.3   Ecological niches for picoplankton groups

By using non-parametric kernel density functions, we investigated the overlapping in the ecological niches of the autotrophic and heterotrophic picoplankton subgroups defined by using the three variables previously considered together with surface

nitrate concentration (Figure 5). Photic layer depth-integrated biomass of each picoplankton group was used for this analysis. These results revealed three ecological niches in the distribution of picoplankton subgroups. *Prochlorococcus* and LNA prokaryotes were more abundant in warm waters, where nitrate supply was low. HNA prokaryotes and *Synechococcus* dominated in cooler regions with medium to high nitrate supply, and picoeukaryotes were more abundant in cold waters with high nitrate supply. A large degree of overlapping of the ecological niches for all picoplankton subgroups was observed when only

surface light was considered. For each picoplankton subgroup Table 4 shows the partial weighted overlap of the ecological niches defined by the four factors: SST, PAR, nitrate flux and surface nitrate concentration. According to these data only nitrate supply enabled a statistically significant separation of the niches of both heterotrophic (HNA and LNA prokaryotes), and autotrophic (*Prochlorococcus*, Synechococcus and picoeukaryotes) picoplankton subgroups. Although the minimum overlap between *Prochlorococcus* and the other autotrophic picoplankton subgroups was also well defined by temperature, only nitrate

supply could statistically distinguish the niche partitioning between the two groups of heterotrophic prokaryotes ($p<0.05$) and between *Synechococcus* and picoeukariotes ($p<0.1$).

## 4   Discussion

### 4.1   Environmental factors and ecological niches

Picoplankton community composition and concurrent estimates of nitrate supply into the euphotic zone from highly contrasting marine environments allowed us to conclude that sea surface temperature and nitrate supply are the main factors controlling the variability in the biomass of different subgroups, whereas surface light generally played a minor role. As far as we know, only one study had previously investigated the role of these environmental factors in the distribution of, in this case, the two major groups of cyanobacteria. By using a large flow cytometry dataset from all major ocean regions, Flombaum et al. (2013)

concluded that temperature and light were the most important predictors of the abundances of *Prochlorococcus* and *Synechococcus*, with nitrate availability exerting a negligible effect. Although this conclusion seems to be contradictory with the





results presented here, some important differences between these studies should be noted. Firstly, Flombaum et al. (2013) used bulk estimates of seawater nitrate concentration as a proxy for nitrate availability in the euphotic zone. However, in near steady-state systems such as the subtropical gyres, where diffusive nutrient supply into the euphotic zone is slow, nitrate concentrations are kept close to the detection limit due to phytoplankton uptake. For this reason, nitrate concentrations and actual nitrate supply into the euphotic zone in the vast oligotrophic regions are often largely disconnected Mouriño-Carballido et al. (2011, 2016). Moreover, whereas Flombaum et al. (2013) used *Prochlorococcus* and *Synechococcus* abundances determined at several depths in the upper 200 m, we used depth-integrated biomass of both autotrophic and heterotrophic picoplankton subgroups in the photic layer.

Although our results point to both temperature and nitrate supply as important factors controlling the distribution of the picoplankton subgroups, nitrate supply was the only factor that allowed the distinction between the ecological niches of autotrophic and heterotrophic picoplankton subgroups. Our attempt to sort out the ecological niches of picoplankton subgroups gave rise to three distinct categories. *Prochlorococcus* and LNA prokaryotes were more abundant in warmer waters (above 20ºC) where the availability of nitrate was low. *Synechococcus* and HNA prokaryotes prevailed mainly in cooler (below 20ºC) marine environments characterized by intermediate and high levels of nitrate supply, and finally, the niche for picoeukaryotes was characterized by lower temperatures and high nitrate supply. These results underline the physiological and ecological features of the distinct picoplankton functional subgroups. Our results confirm the previously reported ecological differences between the two major groups of unicellular cyanobacteria (Scanlan and West, 2002; Partensky and Garczarek, 2010; Li, 2009). Moreover, the ecological niche alignment of the two cyanobacteria genera with the two heterotrophic prokaryotes subgroups is consistent with taxa that prevail in oligotrophic regions (e.g. SAR11) being included in the LNA prokaryotes, whereas copiotrophic and more diverse taxa (Gammaproteobacteria, Bacteroidetes/Flavobacteria, etc) are generally grouped under HNA (Vila-Costa et al., 2012; Schattenhofer et al., 2011). Although the relationship between stratification, mixing and nutrient supply is not obvious (Mouriño-Carballido et al., 2016), our results are in general consistent with the patterns described by Bouman et al. (2011). These authors, by using vertical density stratification as a proxy for the three main environmental factors influencing phytoplankton growth (temperature, light and nutrients) in subtropical regions of the Pacific, Atlantic and Indian Oceans, described the dominance of photosynthetic picoeukaryotes in well-mixed waters, and the prevalence of cyanobacteria in strongly stratified conditions.

## 4.2 Physiological traits of picoplankton subgroups

Although previous studies have revealed that *Prochlorococcus* may have acquired the ability to use nitrate by horizontal gene transfer, their photosynthetic activity primarily relies on regenerated forms of nitrogen (Moore et al., 2002; Malmstrom et al., 2013). Our results support this view and substantiate that, after controlling for the concurrent effects of light and seawater temperature, *Prochlorococcus* biomass was uncorrelated with nitrate fluxes. However, it is important to note that we could not discriminate between high-light (HL) and low-light (LL) ecotypes, and that the presence of nitrate reductase seems to be more relevant in LL (Martiny et al., 2009; Berube et al., 2014). Evolutionary adaptation to light limiting conditions makes





*Prochlorococcus* the most efficient light harvesters among Earth's photosynthetic organisms (Morel et al., 1993). Their competitive ability under light limiting conditions could explain the negative effect of light as a predictor for *Prochlorococcus* biomass. Ultimately, the photo-physiological strategy of *Prochlorococcus* leads to i) thermal sensitivity of photosystem II (Mackey et al., 2013), which limits its fundamental niche to temperatures greater than 15 °C (Moore et al., 1995), and ii) high sensitivity to

5 ultraviolet (UV) radiation in surface waters (Llabrés et al., 2010; Mackey et al., 2013; Sommaruga et al., 2005). This could explain that, after removing the effect of light, our data analysis revealed that the effect of temperature on *Prochlorococcus* biomass showed a sigmoid relationship as temperature increased.

*Synechococcus* is able to use both new and regenerated forms of nitrogen (Moore et al., 2002; Mulholland and Lomas, 2008), which largely explains its wider geographical distribution range (Flombaum et al., 2013). The fact that it is more abundant

at intermediate levels of nitrate supply is consistent with the lower intracellular nitrogen quota of *Synechococcus* relative to *Prochlorococcus*, and hence their higher growth rate under saturating nutrient conditions (Marañón et al., 2013). On the other hand, the large affinity of *Prochlorococcus* to acquire nutrients (Partensky and Garczarek, 2010) and absorb light under severe nutrient- and light-limiting conditions (Mella-Flores et al., 2012), preclude the supremacy of *Synechococcus* in warm and stratified oligotrophic systems (Moore et al., 2007). Although *Prochlorococcus* and *Synechococcus* are not very different in cell size

and they usually coexist in oligotrophic regions, differences in adaptation to light conditions and UV-stress lead to segregate their maximal distributions across space (vertical segregation) and through time (Chisholm, 1992; Mella-Flores et al., 2012).

Picoeukaryotes, like *Synechococcus*, also exhibited a negative relationship with seawater temperature, again reflecting the superior competitive ability of *Prochlorococcus* under severe nutrient-limiting conditions (Moore et al., 2007). The relative dominance of cyanobacteria in oligotrophic systems results from the fact that cyanobacteria are less negatively affected by nu-

20 trient diffusion limitation than picoeukaryotes (Chisholm, 1992). It is widely accepted that small cells are at an advantage over large cells in stratified open ocean environments, where nutrient recycling dominates biogeochemical fluxes (Raven, 1998). First, the surface-to-volume ratio increases with decreasing cell size, which narrows the nutrient diffusion boundary layer around the cell and facilitates the acquisition of nutrients in nutrient impoverished environments. Second, small-sized cells have lower sinking rates than their larger counterparts, which allow them to extend their chances of survival in the euphotic

layer (Smayda, 1980; Chisholm, 1992; Kiørboe, 1993). Our analysis indicates that among the picophytoplankton, picoeukaryotes were the most responsive to nutrient fluxes. This is consistent with experimental observations under laboratory-controlled conditions revealing that, within the picoplankton size range, the maximum attainable growth rate increases with increasing cell size (Raven, 1994; Marañón et al., 2013). This positive relationship between maximum growth rate and cell size in the pico- to small nano-phytoplankton size range has been explained as a trade-off between intracellular nitrogen quotas (N re-

quirements) and mass-specific nitrate uptake rates (N uptake) (Marañón et al., 2013). Whereas nitrogen uptake rate exhibits an isometric relationship with cell size, smaller picoplankton cells have substantially larger intracellular nitrogen quotas, which reduce their capability to maximize carbon-specific growth rates. On the other hand high maximum growth rates represent an advantage for picoeukaryotes, as to any other organism, as they attenuate the effect of loss processes such as predation or the wash-out of plankton communities in highly dynamic, turbulent systems (Sherr et al., 2005; Echevarría et al., 2009). For

instance, the microzooplankton is thought to maintain the biomass of their prey under tight control, and thus slight variations




in picophytoplankton growth rate may substantially alter the resulting biomass of the different picophytoplankton subgroups (Chen et al., 2009).

The unimodal relationship observed between the biomass of picoeukaryotes and nitrate supply could seem at first contradictory with the rising hypothesis proposed by Barber and Hiscock (2006), which describes that improved growth conditions bene-

5 fit all phytoplankton size-classes, including picoplankton. In this regard, Brewin et al. (2014) by using data collected along the Atlantic Meridional Transect cruises showed that <2 μm size-fractionated chlorophyll was positively correlated with total chlorophyll only until a value of 1 mg m$^{-3}$, and then it did not show any positive relationship with total chlorophyll. It is also important to note that surface abundance of picoplankton subgroups reported in our study, which are consistent with previous observations Zubkov et al. (2000); Frojan et al. (2014); Teira et al. (2015), did show higher surface abundance of picoeukary-

10 otes in the Galicia coastal upwelling and the NW Mediterranean compared to the tropical and subtropical Atlantic (Table 2). However, this pattern was diluted when depth-integrated biomasses were computed as the lower limit for the integration (the base of the photic zone) was much shallower in the coastal upwelling domain (ca. 37 m), compared to the NW Mediterranean (ca. 62 m) and the tropical and subtropical regions (ca. 109 m).

Heterotrophic prokaryotes also use dissolved inorganic nutrients, including nitrate, for growth (Kirchman, 2000). Consistent

with this, Gasol et al. (2009) showed a positive relationship between prokaryotic abundance and a proxy for nutrient supply in a latitudinal gradient across the Atlantic. They did not partition this effect on the two subgroups that can universally be differentiated among bacteria and archaea. Our results suggest that LNA prokaryotes respond less markedly to nutrient fluxes than HNA prokaryotes. The effect of nitrate supply on the biomass of LNA prokaryotes showed a linear negative relationship as nitrate supply increases, perhaps associated with their ability to survive under nutrient starving conditions (Mary et al.

2008). Under such conditions, proteorhodopsin-containing LNA prokaryotes (e.g. example SAR11) can use energy from light (Mary et al. 2008; Pinhassi et al. 2016) improving their competitiveness against non-proteorhodopsin-containing prokaryotes. Consistent with this idea, our results showed a positive relationship between the biomass of LNA prokaryotes and PAR. Li et al. (2004) already proposed the ubiquity of this bottom-up control of prokaryotic abundance in oligotrophic environments (<1 mg Chl m$^3$). Therefore, we believe that it is more the difference in the suite of genes (Schattenhofer et al., 2011) than in

cell size (Morán et al., 2015) the underlying cause for the clear niche difference between LNA and HNA prokaryotes.

## 5 Outlook

Picoplankton often dominate marine phytoplankton biomass and primary production in oligotrophic regions (Chisholm 1992; Agawin et al. 2000), contribute overwhelmingly to the recycling of organic matter (Azam et al., 1983; Fenchel, 2008), and

30 could have a substantial contribution to the export of carbon to the deep ocean (Richardson and Jackson, 2007). However, our limited understanding about the factors that control picoplankton community composition constrains our ability to include them in ocean biogeochemical models, and predict the consequences of future global change scenarios. For the first time, by combining observations which allowed us to estimate vertical nutrient fluxes, instead of nitrate concentrations, we investigated



the role of temperature, light and nitrate supply in the distribution of the major autotrophic and heterotrophic picoplankton subgroups. Our results highlight the role of nitrate supply in the distribution of picoplankton subgroups, as it was the only factor that allowed the statistically significant distinction of the ecological niches between the autotrophic and heterotrophic picoplankton subgroups. In general, autotrophic picoplankton biomass was dominated by *Prochlorococcus* in warmer waters

where the availability of nitrate was low, and by *Synechococcus* and picoeukaryotes in cooler waters with medium to high nitrate availability. Similarly, LNA prokaryotes dominated heterotrophic picoplankton biomass in regions of weak nitrate supply, whereas HNA prokaryotes dominated the heterotrophic community in regions of enhanced nutrient supply. Although our study included 97 stations sampled in contrasting environments, a larger dataset, including a broader range of environmental conditions, will be needed to accurately discern the role of temperature and nitrate supply in the field, as both factors are strongly

correlated in the ocean. In this regard, by growing three phytoplankton species (the diatom *Skeletonema costatum*, the coccolithophore *Emiliania huxleyi* and the picocyanobacteria *Synechococcus* spp.) in the lab, Marañón et al. (2018) showed a reduced sensitivity of metabolic rates to temperature variability under nutrient-depleted conditions, suggesting that nutrient availability controls the temperature dependence of metabolism. Consistent with these results, our statistical analyses stress the relevance of nitrate supply in the distinction of the ecological niches of heterotrophic and autotrophic picoplankton subgroups. Other

mechanisms of nutrient supply, such as mesoscale and sub-mesoscale turbulence, atmospheric deposition, nitrogen fixation and more complex 3-dimensional dynamics Jenkins and Doney (2003); Bonnet et al. (2005); Estrada et al. (2014); Fernández-Castro et al. (2015), as well as the influence of trophic interactions Van Mooy et al. (2006); Baudoux et al. (2007); Chen et al. (2009); Rusch et al. (2010) deserve further investigation.

According to our results, in a future ocean where global change scenarios predict an increase in surface temperature and

20 stratification (Howes et al. 2015), the resulting decrease in nitrate supply into the euphotic zone Lewandowska et al. (2014) would lead to the dominance of autotrophic picoplankton by cyanobacteria, whereas the picoeukaryotes would decrease their contribution. In order to get a rough estimation of this effect, we computed the cyanobacteria to picoeukaryotic biomass ratio for the present and a future global change scenario (See supplementary Figure A2 and Methods). According to this analysis, the projected decrease in averaged global nitrate supply (ca. 20%) would lead to an average increase of 9% in the ratio of

25 cyanobacteria to picoeukaryotes. Due to the smaller contribution of cyanobacteria to the transfer of carbon to the deep ocean compared to picoeukaryotes, this pattern could have important implications regarding the contribution of the smaller cells to the biological carbon pump (Corno et al., 2007).

*Author contributions.* JO., PC and BM. designed the research; JO. analyzed the data and JO. and BM. prepared the manuscript with contri-

30 butions from all the co-authors.

*Competing interests.* The authors declare no conflict of interest.





*Acknowledgements.* We thank the officers and crew of the R/V Hespérides, Sarmiento de Gamboa, Mytilus, Ramon Margalef and Lura for their help during the cruises. We are also very grateful to Fátima Eiroa for the flow cytometry analysis in NICANOR, HERCULES1, HERCULES2, HERCULES3 and ASIMUTH cruises. This research was supported by the Spanish Ministry of Economy and Competitiveness (MINECO) through projects: CTM2012-30680 to Beatriz Mouriño, CTM2008-0626I-C03-01 to Mikel Latasa, REN2003-09532-C03-01 to

5   Ramiro Varela Benvenuto, CTM2004-05174-C02 to Emilio Marañón, CTM2011-25035 to Pedro Cermeño, 09MMA027604PR to Manuel Ruiz Villareal and from the Galician Goverment through grant EM2013/021 to Beatriz Mouriño. Jose Luis Otero-Ferrer acknowledges the receipt of a FPI contract from MINECO (CTM2012-30680), and Bieito Fernández Castro a Juan de La Cierva Formación fellowship (FJCI-641 2015-25712, Ministerio de Economía y Competitividad, Spanish Goverment).





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



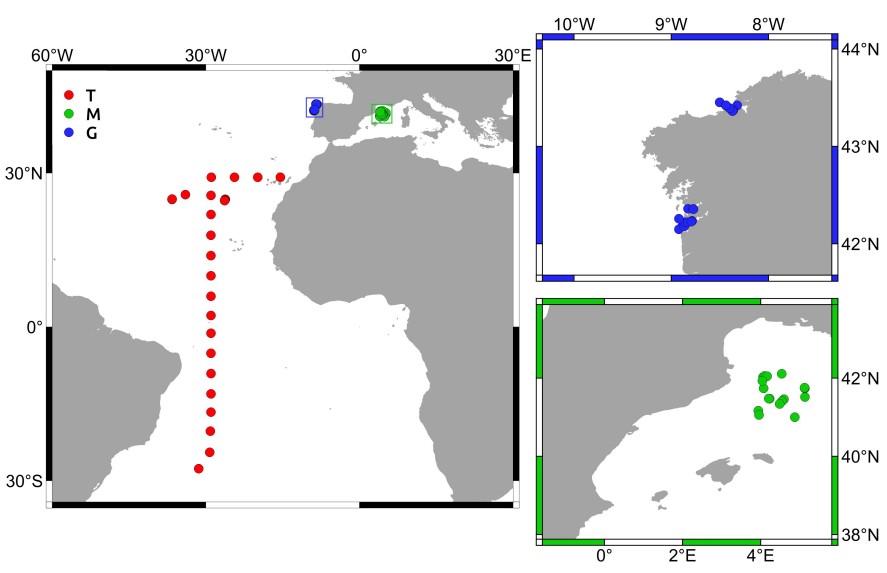

**Figure 1.** Location of the stations sampled in the tropical and subtropical Atlantic ocean (T), the northwestern Mediterranean Sea (M), and the Galician coastal upwelling (G). Small panels provide details about those staions sampled in M (green color) and G (blue color).





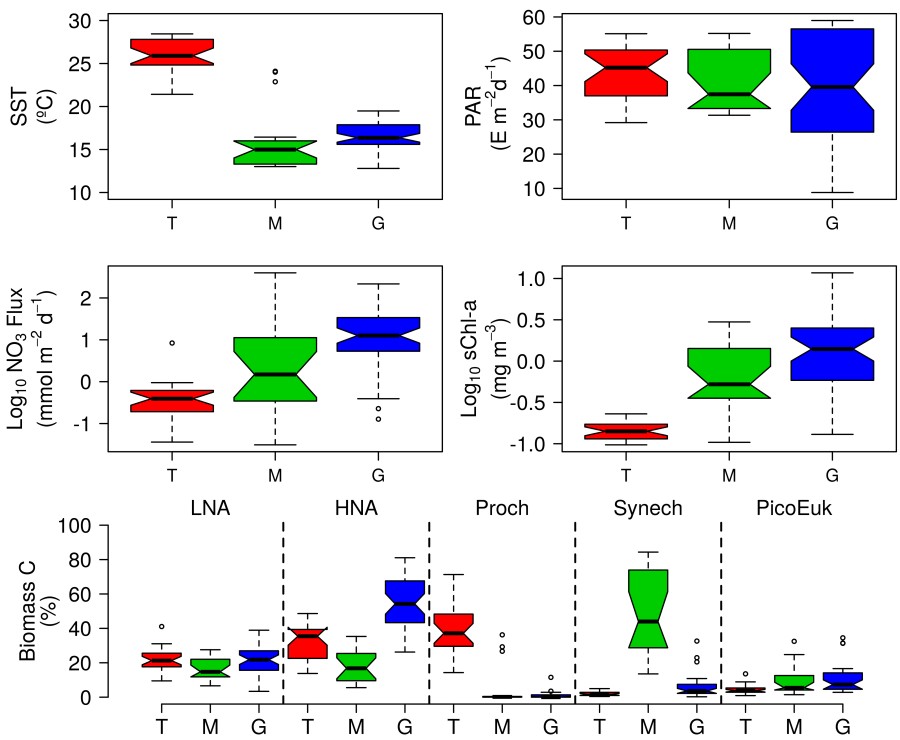

**Figure 2.** Box-and-whiskers plots of Sea Surface Temperature (SST), surface photosynthetic active radiation (PAR), nitrate supply (NO3 Flux), surface chlorophyll-*a* concentration (sChl-a), and contribution to total picoplankton biomass of low (LNA) and high (HNA) nucleic acid content bacteria, *Prochlorococcus* (Proch), *Synechococcus* (Synech), and small picoeukaryotes (PicoEuk) computed for the tropical and subtropical Atlantic ocean (T), the Northwestern Mediterranean (M) and the Galician coastal upwelling (G). On each box, the central mark indicates the median, the notches the 95% confidence interval for the median, and the bottom and top edges of the box indicate the 25[th] and 75[th] percentiles, respectively. The whiskers extend to the most extreme data points not considered outliers, and the outliers are plotted individually using white circles.





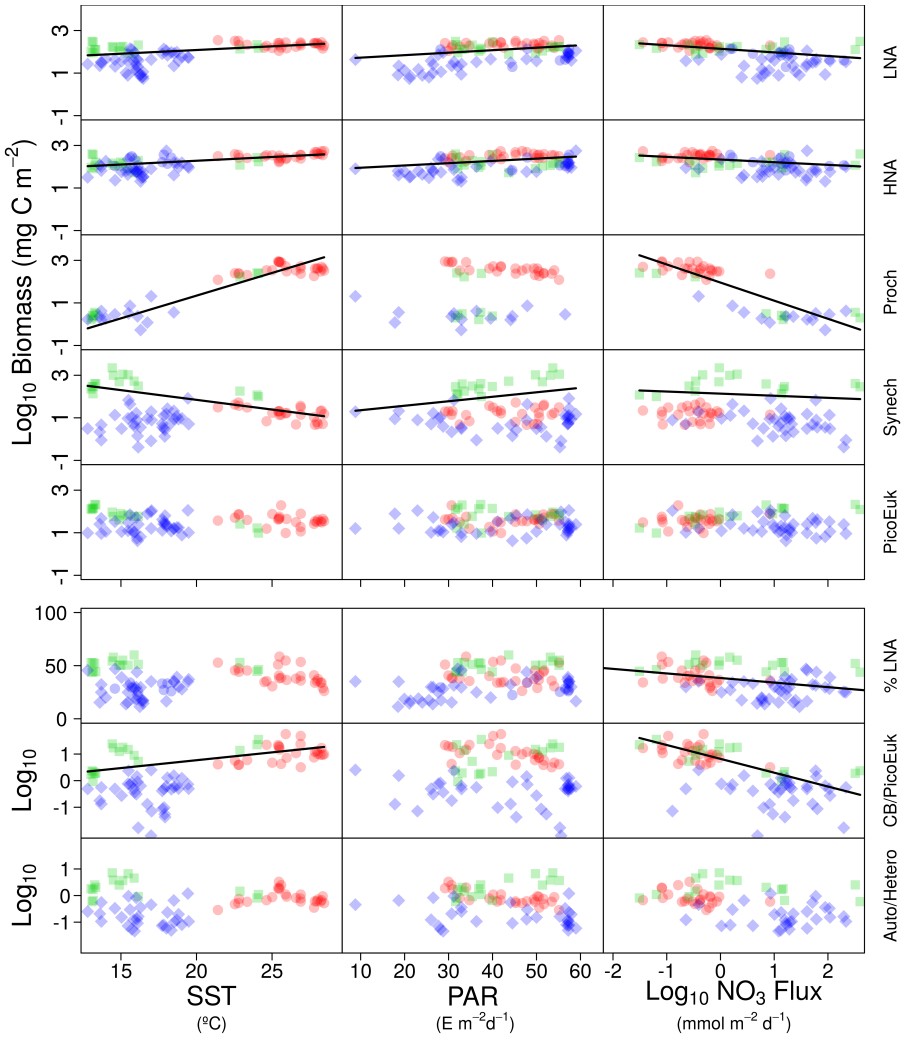

**Figure 3.** Pair scatter plots representing the relationship between log-transformed depth-integrated biomass for each picoplankton subgroup, the contribution of low nucleic acid content bacteria to heterotrophic picoplankton biomass (%LNA), the ratio of cyanobacteria (*Prochloro-coccus* + *Synechococcus*) to picoeukaryotes depth-integrated biomass (CB/PicoEuK), and the ratio autotrophic to heterotrophic picoplankton biomass (Auto/Hetero) versus sea surface temperature (SST), surface photosynthetically active radiation (PAR) and nitrate flux (NO3 flux). Significant linear relationships are indicated as solid (p-value < 0.01) black lines. Samples collected at different regions are indicated as red dots (tropical and subtropical Atlantic ocean), green squares (NW Mediterranean) and blue diamonds (Galician coastal upwelling).



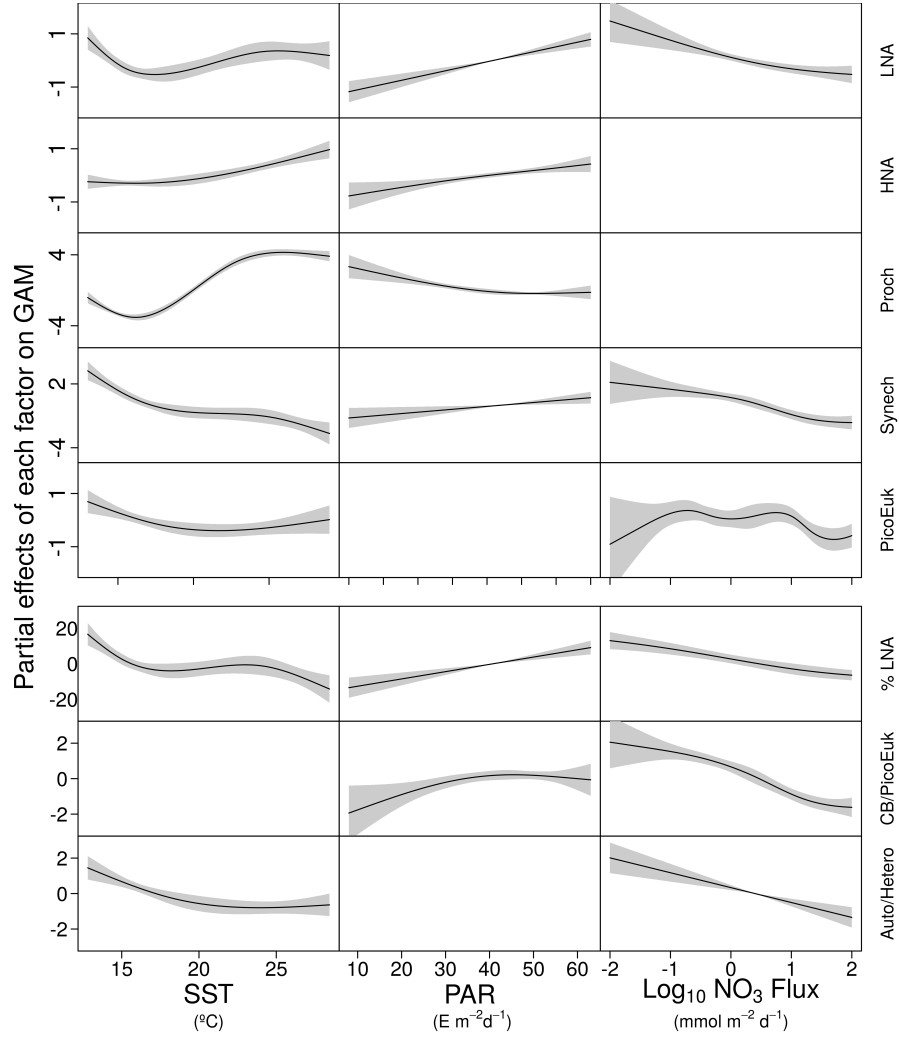

**Figure 4.** GAM-predicted effects of the response variables (biomass and contribution of picoplankton subgroups) as a smooth function of surface temperature (SST), photosynthetically active radiation (PAR) and nitrate flux (NO$_3$ flux). All terms were centered in zero. Significant linear relationships are indicated as solid (p-value < 0.01) black lines. Shaded regions represent the 95% confidence intervals of the smooth spline functions. Intercept values were 4.6 (LNA), 5.1 (HNA), 2.1 (*Prochlorococcus*), 3.6 (*Synechococcus*), 3.7 (picoeukaryotes), 36.4 (contribution of LNA to heterotrophic picoplankton, %LNA), 1.4 (cyanobacteria to picoeukaryotes ratio, CB/PicoEuk), and -0.1 (autotrophic to heterotrophic biomass ratio, Auto/Hetero).





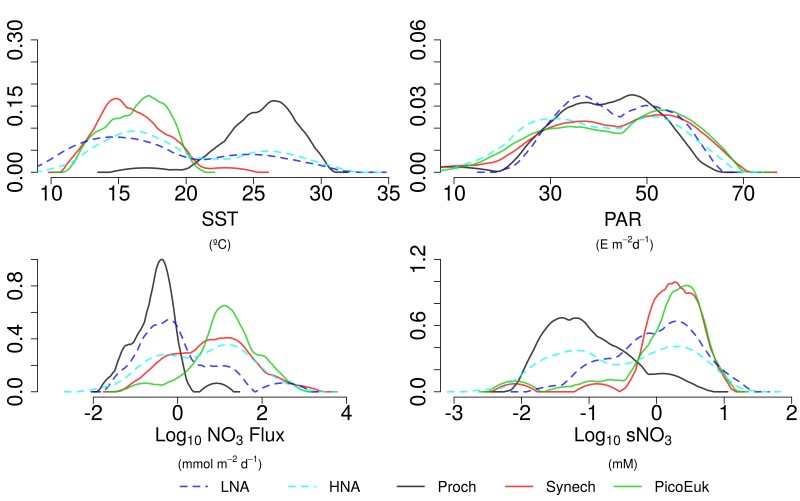

**Figure 5.** Kernel density estimates of LNA and HNA bacteria, *Prochlorococcus*, *Synechococcus* and picoeukaryotes based on the considered niche descriptors: sea surface temperature (SST), surface photosynthetically active radiation (PAR), nitrate flux (NO$_3$ flux) and surface concentration ( sNO$_3$).





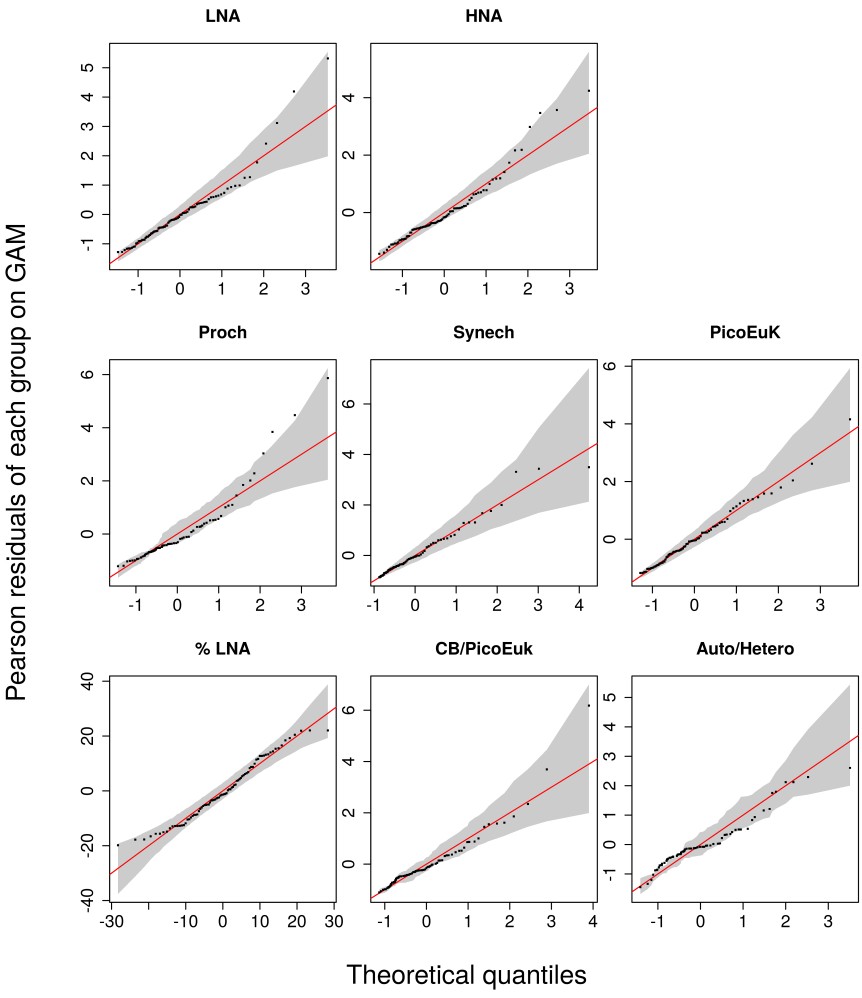

**Figure A1.** Quantile-Quantile (QQ) plots between the observations and the selected GAM models for each picoplankton subgroup, the contribution of LNA to heterotrophic picoplankton (%LNA), the cyanobacteria to picoeukaryotes ratio (CB/PicoEuk), and the autotrophic to heterotrophic biomass ratio (Auto/Hetero). The y-axes represent the Pearson residuals and the x-axes the negative binomial theoretical quantiles. Solid red lines indicate the theoretical quantile of the models and grey shadows the 95% confidence intervals.



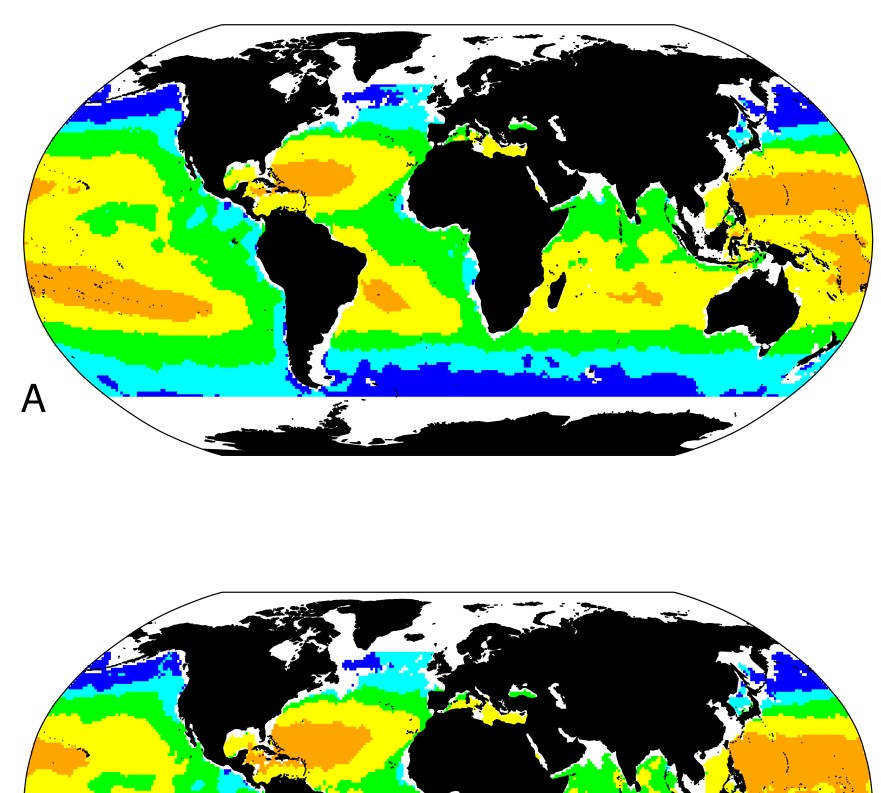

**Figure A2.** Present (A) and future (2100) (B) distributions of the cyanobacteria to picoeukaryotes depth-integrated biomass ratio (see Methods). Regions in white color represents data that was outside of the range covered by the dataset used to build our prediction model.



**Table 1.** Details of the data included in this study. Domain referred to the tropical and subtropical Atlantic ocean (T), the Northwestern Mediterranean Sea (M), and the Galician coastal upwelling (G). N indicates the number of stations sampled at each cruise

| Domain | Region | N | Cruise | Vessel | Date |
|--------|--------|---|--------|--------|------|
| T | NE Atlantic | 8 | CARPOS | Hespérides | 14/10/06- 22/11/06 |
| T | Atlantic | 18 | TRYNITROP | Hespérides | 14/04/08 - 02/05/08 |
| M | Liguro-Provençal Basin | 6 | FAMOSO I | Sarmiento de Gamboa | 14/3/09 - 22/3/09 |
| M | Liguro-Provençal Basin | 10 | FAMOSO II | Sarmiento de Gamboa | 30/4/09 - 13/05/09 |
| M | Liguro-Provençal Basin | 3 | FAMOSO III | Sarmiento de Gamboa | 16/09/09 - 20/09/09 |
| G | Ría de A Coruña | 1 | HERCULES I | Lura | 07/06/10 |
| G | Ría de A Coruña | 5 | HERCULES II | Lura | 28/09/11 - 29/09/11 |
| G | Ría de A Coruña | 13 | HERCULES III | Lura | 16/07/12 - 20/07/12 |
| G | Ría de Vigo | 9 | DISTRAL | Mytilus | 14/02/12 - 06/11/12 |
| G | Ría de Vigo | 2 | CHAOS | Mytilus | 20/08/13 - 27/08/13 |
| G | Ría de A Coruña | 12 | NICANOR | Lura | 27/02/14 - 17/12/15 |
| G | Rías de Vigo & Pontevedra | 10 | ASIMUTH | Ramón Margalef | 17/06/13 - 21/06/13 |





**Table 2.** Mean ± standard deviation of Sea Surface Temperature (SST), surface Photosynthetic Active Radiation (PAR), Mixed Layer Depth (MLD), Photic Layer Depth (1% PAR), surface nitrate concentration (sNO₃), nitrate gradient, vertical diffusivity (K), nitrate supply (NO₃ flux), surface chlorophyll (sChl-*a*), photic layer depth-integrated chlorophyll-a (Chl-*a*), biomass (B), abundance (A), and contribution (C) to total picoplankton biomass (Total Pico B), and surface abundance (s) of LNA and HNA bacteria, *Prochlorococcus*, *Synechococcus* and picoeukaryotes computed for the Tropical and subtropical Atlantic ocean (T), the Northwestern Mediterranean (M), and the Galician coastal upwelling (G). MLD was estimated from an increase in water column density of 0.125 Kg m⁻³ relative to surface values. A nonparametric 1-way ANOVA (Kruskal-Wallis) was performed to test the null hypothesis that independent groups come from same distribution. The Bonferroni multiple comparison test was applied a posteriori to analyze the differences between every pair of groups. *p < 0.05; **p < 0.01; ***p < 0.001.

| Variables (Units) | T | M | G | KW p-value | Post hoc *Bonferroni* |
|---|---|---|---|---|---|
| SST (ºC) | 26 ± 2 | 16 ± 4 | 16 ± 2 | <0.001*** | T>G>M |
| PAR (E m⁻² d⁻¹) | 43 ± 23 | 42 ± 13 | 39 ± 17 | 0.69 | |
| MLD (m) | 61 ± 30 | 61 ± 71 | 12 ± 10 | <0.001** | T,M>G |
| 1% PAR (m) | 109 ± 23 | 62 ± 13 | 37 ± 17 | <0.001** | T>M>G |
| sNO₃ (µmol m⁻³) | 90 ± 77 | 2414± 1635 | 1601 ± 1604 | <0.001*** | T<M,G |
| NO₃ gradient (µmol m⁻⁴) | 146 ± 158 | 90 ± 40 | 105 ± 100 | 0.71 | |
| K (m² s⁻¹) x10⁻³ | 0.2 ± 0.3 | 5.3 ± 13.9 | 0.5 ± 0.8 | 0.24 | |
| NO₃ flux (mmol m⁻² d⁻¹) | 0.7 ± 1.6 | 41 ± 113 | 30 ± 46 | <0.001*** | T<M<G |
| sChl-*a* (mg m⁻³) | 0.1 ± 0.1 | 0.9 ± 0.9 | 2.2 ± 2.5 | <0.01** | T<M<G |
| Chl-*a* (mg m⁻²) | 31 ± 6 | 70 ± 99 | 81 ± 66 | <0.001*** | T<G |
| sLNA A (cell ml⁻¹) x105 | 4.4±2.4 | 3.7±2.5 | 2.1±1.1 | <0.001*** | T>M>G |
| sHNA A (cell ml⁻¹) x105 | 3.0±1.8 | 4.0±4.5 | 3.6±2.3 | 0.13 | |
| sProchl A (cell ml⁻¹) x10³ | 144±132 | 2.2±4.4 | 1.0±2.8 | <0.001*** | T>M,G |
| sSynech A (cell ml⁻¹) x10³ | 18±66 | 75±81 | 5.7±6.9 | <0.001*** | T<M>G |
| sPicoEuk A (cell ml⁻¹) x10³ | 2.5±9.4 | 6.8±8.4 | 5.7±6.9 | <0.001*** | T<M,G |
| LNA A (cell m⁻²) x10¹² | 40 ± 20 | 22 ± 8 | 6.4 ± 4 | <0.001*** | T,M>G |
| HNA A (cell m⁻²) x10¹² | 27 ± 1 | 22 ± 1 | 9.4 ± 0.8 | <0.001*** | T>M>G |
| Prochl A (cell m⁻²) x10¹¹ | 156 ± 121 | 10 ± 23 | 0.5 ± 1 | <0.001*** | T>M,G |
| Synech A (cell m⁻²) x10¹¹ | 7 ± 15 | 50 ± 49 | 2 ± 2 | <0.001*** | T<M>G |
| PicoEuk A (cell m⁻²) x10¹¹ | 1.7 ± 3 | 2.8 ± 2 | 1 ± 2 | <0.001*** | T<M>G |
| LNA B (mg C m⁻²) | 253 ± 105 | 170 ± 97 | 43 ± 34 | <0.001*** | T>M>G |
| HNA B (mg C m⁻²) | 216 ± 127 | 168 ± 105 | 108 ± 73 | 0.02* | T>M>G |
| Prochl B (mg C m⁻²) | 482 ± 516 | 36 ± 84 | 1.3 ± 4 | <0.001*** | T>M,G |
| Synech B (mg C m⁻²) | 43 ± 83 | 576 ± 530 | 19 ± 26 | <0.001*** | T,M>G |
| PicoEuk B (mg C m⁻²) | 59 ± 102 | 86 ± 59 | 43 ± 59 | <0.001*** | T<M >G |
| Total Pico B (mg C m⁻²) | 1052 ± 215 | 1038 ± 485 | 216 ± 36 | <0.001*** | T,M>G |
| LNA C (%) | 27 ± 10 | 18 ± 8 | 21 ± 9 | <0.001*** | T>M,G |
| HNA C (%) | 22 ± 12 | 17 ± 10 | 55 ± 15 | <0.001*** | T,M<G |
| Prochl C (%) | 41 ± 16 | 5 ± 12 | 1 ± 2 | <0.001*** | T,M<G |
| Synech C (%) | 4 ± 5 | 50 ± 24 | 6 ± 7 | <0.001*** | T,G>M |
| PicoEuk C (%) | 5 ± 5 | 10 ± 9 | 11 ± 9 | <0.001*** | T<M,G |





**Table 3.** Simple ($R^2$) and adjusted squared correlation coefficients (Adj-$R^2$) for simple linear regression and multiple Generalized Additive Models (GAM) built to predict depth-integrated biomass for each picoplankton subgroup, the contribution of LNA bacteria to total heterotrophic picoplankton biomass (% LNA), the ratio cyanobacteria (*Prochlorococcus* + *Synechococcus*) to picoeukaryotes depth-integrated biomass (CB/PicoEuK) and the ratio autotrophic (CB+PicoEuk) to heterotrophic bacteria (LNA + HNA) biomass based on Sea Surface Temperature (SST), surface photosynthetically active radiation (PAR) and nitrate supply (NO$_3$ flux). Negative binomial distribution was assumed. Multiple model selection was based on stepwise regression and the Akaike Information Criterion (see Methods). Only significant (p-value $< 0.05$) results are shown. Percentage of total effects represent the contribution of each environmental factor to the variability explained by each GAM model (see methods).

| Group | $R^2$ simple linear | | | Adj-$R^2$ multiple regression | Percentage of total effects | | |
|---|---|---|---|---|---|---|---|
| | SST | PAR | NO$_3$ flux | | SST | PAR | NO$_3$ flux |
| LNA | 0.39 | <0.01 | 0.34 | 0.55 | 0.30 | 0.35 | 0.35 |
| HNA | 0.47 | 0.05 | 0.18 | 0.53 | 0.52 | 0.48 | |
| Proch | <0.01 | | <0.01 | 0.86 | 0.72 | 0.28 | |
| Synech | 0.11 | <0.01 | | 0.52 | 0.53 | 0.16 | 0.31 |
| PicoEuk | | | | 0.23 | 0.51 | | 0.49 |
| % LNA | | 0.05 | 0.12 | 0.49 | 0.39 | 0.26 | 0.35 |
| CB/PicoEuk | 0.25 | | 0.28 | 0.40 | | 0.38 | 0.62 |
| Auto/Hetero | | | | 0.29 | 0.39 | | 0.61 |





**Table 4.** Partial weighted niche overlap (%) for each environmental factor and picoplankton subgroup. $sNO_3$ represent surface nitrate concentration. Asterisks denote the existence of significant differences between niches (^ $p<0.1$, * $p<0.05$, ** $p<0.01$, *** $p<0.001$).

|          |         | LNA | HNA | Proch | Synech | PicoEuk |
|----------|---------|-----|-----|-------|--------|---------|
|          | LNA     | 100 |     |       |        |         |
|          | HNA     | 85  | 100 |       |        |         |
| **SST**  | Proch   |     |     | 100   |        |         |
|          | Synech  |     |     | 9***  | 100    |         |
|          | PicoEuk |     |     | 5***  | 84     | 100     |
|          | LNA     | 100 |     |       |        |         |
|          | HNA     | 80  | 100 |       |        |         |
| **PAR**  | Proch   |     |     | 100   |        |         |
|          | Synech  |     |     | 80    | 100    |         |
|          | PicoEuk |     |     | 74*   | 94     | 100     |
|          | LNA     | 100 |     |       |        |         |
|          | HNA     | 69* | 100 |       |        |         |
| **NO₃ Flux** | Proch |     |     | 100   |        |         |
|          | Synech  |     |     | 31*** | 100    |         |
|          | PicoEuk |     |     | 14*** | 77 ^   | 100     |
|          | LNA     | 100 |     |       |        |         |
|          | HNA     | 73^ | 100 |       |        |         |
| **sNO₃** | Proch   |     |     | 100   |        |         |
|          | Synech  |     |     | 22*** | 100    |         |
|          | PicoEuk |     |     | 29*** | 89     | 100     |