# Peer review of "Factors controlling the community structure of picoplankton in contrasting marine environments"

_Biogeosciences, 2018_

## Referee Comment (RC1) · Anonymous Referee #1 · 9 Jul 2018

The manuscript investigates the relationship between nutrient supply estimated by measurements of turbulent mixing and nitrate gradients and the community structure of picophytoplankton (including both autotrophic and heterotrophic groups). The take-home message of the work is that studies that use ambient nutrient concentrations as a proxy of nutrient availability could be misleading as in many regions of the ocean the supply of nutrients by turbulent diffusion is not registered in bottle samples of nutrients. This is an important message that needs to be communicated to the wider marine science community.

The manuscript is well written and provides a nice overview of the ecological literature

of marine picophytoplankton. The dataset of turbulent mixing, nitrate concentrations and picoplankton community structure is novel and covers a variety of hydrographic and trophic regimes.

I have a few questions and comments concerning both the estimation of nutrient supply using combined MSS and nutrient profiles and the choice of sampling stations used in the analysis which I hope the authors may be able to address.

NUTRIENT FLUX ESTIMATES Although the authors correctly point out that concomitant datasets of turbulent mixing and picophytoplankton community structure are rare, this may be due in part to the lack of high-quality estimates of nutrient flux from microstructure profile measurements in the surface ocean with coincident depth-resolved nutrient profiles (required to obtain a robust estimate of the nutrient gradient near the nutricline). The vertical resolution of the nutrient profiles within the dataset is unclear (only a range between 3 and 9 depths is provided, but it could be tricky to use 3/4 depths to provide a good estimate of the nutrient gradient ). Could the extent of the density gradient be over/underestimated in cases where the depth resolution is low and by how much? It would be good to have a frequency histogram for the dataset showing the number of depths per profile so the reader is aware of the vertical resolution of nutrient concentrations across the dataset.

DENSITY AND NITRATE RELATIONSHIPS The authors also mention that for some of the stations nutrient data was not available, and instead of nutrient bottle data, a relationship between density and nitrate was used. Again it is not clear how robust the relationship between nitrate concentration and density was for the relevant stations. Could the authors provide supplementary plots of the nutrient versus density relationship that was used to estimate nitrate gradients, similar to that of Williams et al. (2013a GRL 40:5467-5472; 2013b Limnol. & Oceanogr.: Fluids and Envs 3:61-73)?

EPISODIC NATURE OF MIXING Mixing events in some regions can be episodic, yet short-term vertical pulses of nutrients can trigger significant shifts in community structure. In some oceanic regions tidal mixing can also be important. The authors mention that 3-10 profiles were taken, but it would also be helpful to know the time interval over which these profiles were made and how this varied between the three study regions (again a frequency histogram documenting this would be helpful). Could it be that for some regions the flux of nutrients could be significantly underestimated given that such short-term events may not be captured in MSS profile data? Given the general audience of the journal, both the strengths and shortcomings of this method of estimating nutrient supply should be provided.

PICOPHYTOPLANKTON BIOMASS The authors report the estimates of picophytoplankton biomass and ratios of biomass, but I was unable to find how the authors convert from cell abundance to carbon per unit volume. This is quite important, as there are several group-specific carbon conversion factors in the literature fand for the larger eukaryotic cells it is likely a biovolume conversion factor may provide a better estimate, as the size range within this subgroup can be significant.

EXTRAPOLATION TO THE GLOBAL OCEAN I was surprised to see that over half of the stations used were from coastal embayments. One could argue from many points of view that these regions may not be representative of the open-ocean eutrophic environments (likely different taxonomic diversity within these gross cytometric groupings, potential supply of nutrients from terrigenous sources, different light environment caused by attenuation by CDOM and sediments, need to correct for advective flux). Perhaps the authors have supporting literature/data that would help convince the reader that these embayments broadly reflect the environmental conditions of offshore stations, but even with such supporting information they should also highlight the need for data from open-ocean meso- and eutrophic environments that would help further resolve the global relationship between mixing and picoplankton community structure.

The dataset is largely confined to a specific geographic region (40N-30S, covering a limited number of biochemical provinces in the Atlantic Basin), yet the authors use relationships from this study to predict future changes in picoplankton community structure

across the entire globe. Would the authors consider limiting their predictions of future community structure to the geographic regions/ latitudinal gradients that are used to develop the predictive models? Given that the global ocean covers a variety of biogeochemical regimes, some of which may not be limited by nitrate, restricting the geographical scope of the future predictions may be advisable, even though the overall patterns tend to broadly resemble those from other global studies.

There is also very limited information on how the global ocean model simulations of nitrogen flux from Lewandowska et al. (2014) were used to estimate percentage change in cyano:picoeuk biomass ratio. Also information on the predictive model setup and assumptions (physical and biogeochemical) would be helpful.

USE OF MOREL MODEL TO ESTIMATE DEPTH OF PHOTIC LAYER Is the light attenuation observed in the Galician coastal stations largely a result of phytoplankton or other optically-active substances? I mention this because the model of Morel used to estimate euphotic depth is restricted to Case-1 (open ocean) waters where light attenuation is dominated by phytoplankton.

---

## Referee Comment (RC2) · Anonymous Referee #2 · 26 Aug 2018

The manuscript by Otero-Ferrer et al reports the relationship between nitrate supply and temperature in the structure of picoplankton groups determined by flow cytometer. The manuscript is really well written, and the literature seems extensively covered. The strength of this work resides in the use of nitrate diffuse flux as a proxy of nutrient availability and depth integrate biomass for the different picoplankton groups to predict the niche of the groups analyzed. Commonly, these measurements are treated as discrete rather than continuous variables. The data and results presented by Otero-Ferrer et al have lot of potential and I am confident that the community will benefit from its publication. However, I believe there are very few points that can be amended in a way to improve clarity of the main message of this work. 1- Plots with the vertical

distribution of the main variables considered (nitrate, cell abundance of picoplankton groups and temperature) could be provided and would certainly help the reader to have a better assessment of the conditions in the sampled stations. 2- Which values of cell-to-carbon conversion factors were used to transform abundance into biomass of the different groups of picoplankton analyzed? 3- The depth integrated biomass of picoeuks was not linearly correlated with temperature, PAR nor nitrate flux when the authors used the simple linear model. However, with the additive model, picoeuks showed a negative trend with temperature and unimodal distribution with nitrate. Could the authors elaborate a bit more in the discussion about this contradiction between methods? 4- The dataset of this manuscript was originated from coastal waters rather than oceanic, from two oceanic regions (Atlantic and Med Sea) and it is confined to a narrow latitude range.Thus It does not support extrapolations to worldwide oceans. I recommend the authors to be more caution and remove figure A2 and the lines 23 to 27 of the last paragraph. SPECIFIC POINTS: INTRO Line 26 – missing a space between the word communities and the reference. MM Section 2, Line 5 – please keep one abbreviation for the Med sea to avoid confusion by writing the only once northwestern. No need to repeat every time since for the Atlantic the same was done. Section 2, line 14 – Diaz et al 2018 does not seem to be on bioxriv or any other repository, thus the info is not available. I would not cite unless the paper has been already released. Section 2.6, line 6 – add Generalized Linear Models before GLM abbreviation since it is the 1st time that appears. RESULTS Section 4.1, line 15, please add the average temperature value for picoeuks, especially because it seems very close to the one found for syn. DISCUSSION The figures and tables still can be cited in the discussion section. It facilitates a lot the follow up of the points discussed.

---

## Author Comment (AC1) · 14 Sep 2018

Anonymous Referee 1 Referee comments shown in black, Author replies shown in blue, Changes to manuscript in purple

Overall comments The manuscript investigates the relationship between nutrient supply estimated by measurements of turbulent mixing and nitrate gradients and the community structure of picophytoplankton (including both autotrophic and heterotrophic groups). The take-home message of the work is that studies that use ambient nutrient concentrations as a proxy of nutrient availability could be misleading as in many regions of the ocean the supply of nutrients by turbulent diffusion is not registered in bottle samples of nutrients. This is an important message that needs to be communicated to the wider marine science community. The manuscript is well written and provides a nice overview of the ecological literature of marine picophytoplankton. The dataset of turbulent mixing, nitrate concentrations and picoplankton community structure is novel and covers a variety of hydrographic and trophic regimes. I have a few questions and comments concerning both the estimation of nutrient supply using combined MSS and nutrient profiles and the choice of sampling stations used in the analysis which I hope the authors may be able to address.

Specific comments 1) NUTRIENT FLUX ESTIMATES Although the authors correctly point out that concomitant datasets of turbulent mixing and picophytoplankton community structure are rare, this may be due in part to the lack of high-quality estimates of nutrient flux from microstructure profile measurements in the surface ocean with coincident depth-resolved nutrient profiles (required to obtain a robust estimate of the nutrient gradient near the nutricline). The vertical resolution of the nutrient profiles within the dataset is unclear (only a range between 3 and 9 depths is provided, but it could be tricky to use 3/4 depths to provide a good estimate of the nutrient gradient ). Could the extent of the density gradient be over/underestimated in cases where the depth resolution is low and by how much? It would be good to have a frequency histogram for the dataset showing the number of depths per profile so the reader is aware of the vertical resolution of nutrient concentrations across the dataset.

We agree with the reviewer that the lower vertical resolution of nitrate concentration in comparison to turbulent data incorporates some uncertainty, which is very hard to evaluate, in the calculation of nitrate diffusive fluxes. Lower vertical resolution ($5\pm2$ sampling depths) was used in the shallower stations sampled in the Galician coastal upwelling, whereas in the deeper NW Mediterranean and the tropical and subtropical regions we used $7\pm1$ and $11\pm2$ depths, respectively. The text has been modified and a new figure (Figure A2.A) has been included in the supplementary material, to represent the frequency histogram of sampling depths used to compute the nitrate

vertical gradient in each region The maximum sampling depth where the microstructure turbulence profiler was deployed is now indicated in Table 1.

Original Page 5 – Line 11-13 Samples for the determination of nitrate (NO3) + nitrite (NO2 ) were collected from 3 to 9 depths in rinsed polyethylene tubes and stored frozen at -20 °C until analysis on land, according to standard methods using the automated colorimetric technique (Grasshoff et al., 2007).

Modified to Page 5 – Line 21-26 Samples for the determination of nitrate (NO3) + nitrite (NO2 ) were collected, on average, from $5\pm2$ (Galician coastal upwelling), $7\pm1$ (NW Mediterranean) and $11\pm2$ (tropical and subtropical Atlantic Ocean) depths in rinsed polyethylene tubes and stored frozen at -20 °C until analysis on land, according to standard methods using the automated colorimetric technique (Grasshoff et al., 2007). The frequency histogram of sampling depths collected for the determination of nitrate concentration in each region is indicated in Figure A2-A in the Supplementary material, whereas the maximum sampling depth where the microstructure turbulence profiler was deployed is indicated in Table 1.

2) DENSITY AND NITRATE RELATIONSHIPS The authors also mention that for some of the stations nutrient data was not available, and instead of nutrient bottle data, a relationship between density and nitrate was used. Again it is not clear how robust the relationship between nitrate concentration and density was for the relevant stations. Could the authors provide supplementary plots of the nutrient versus density relationship that was used to estimate nitrate gradients, similar to that of Williams et al. (2013a GRL 40:5467-5472; 2013b Limnol. Oceanogr.: Fluids and Envs 3:61-73)?

Only in one station sampled in the Galician coastal upwelling ecosystem during the NICANOR project observations of nutrient concentration were not available. Instead nitrate concentration was computed from a nitrate-density (sigma-t) relationship built by using all samples (n=52) collected during the NICANOR sampling period. A new figure (Figure A2-B) has been included in the supplementary material to show this

relationship. The relationship showed a linear behavior (NO−3 = 9.7788 *sigma-t − 256.38; r =0.930; p < 0.001) for density ranging between 26.1 and 27.1 kg m−3.

Original: Page 5 – Line 13-14 In those stations carried out during the NICANOR cruises, where nitrate concentrations were not available, they were obtained by using nitrate-density relationships (Moreira-Coello et al., 2017)

Change: Page 5 – Line 26-29 In one station carried out during the NICANOR cruise, where nitrate concentrations were not available, these were obtained by using a nitrate-density relationship built by using all samples (n=52) collected during the NICANOR sampling period. The relationship showed a linear behavior (NO3 = 9.7788 *$\sigma$t − 256.38; Adj-r2 =0.87; p < 0.001) for density ranging between 26.1 and 27.1 kg m−3 (Figure A2-B).

3) EPISODIC NATURE OF MIXING Mixing events in some regions can be episodic, yet short-term vertical pulses of nutrients can trigger significant shifts in community structure. In some oceanic regions tidal mixing can also be important. The authors mention that 3-10 profiles were taken, but it would also be helpful to know the time interval over which these profiles were made and how this varied between the three study regions (again a frequency histogram documenting this would be helpful). Could it be that for some regions the flux of nutrients could be significantly underestimated given that such short-term events may not be captured in MSS profile data? Given the general audience of the journal, both the strengths and shortcomings of this method of estimating nutrient supply should be provided.

We completely agree with the reviewer that episodic bursts of turbulence can induce episodic inputs of nutrient supply which can be easily missed in sets of 2-10 profiles. The microstructure turbulence profiles used for computing nitrate fluxes at each station were always deployed successively. Sets include 2-11 in the tropical and subtropical Atlantic (37 ± 18 min), 6-7 in the NW Mediterranean (76 ± 22 min) and 3-402 in the Galician coastal upwelling (65 ± 246 min). Our dataset includes two high-frequency

samplings carried out in the outer part of Ría de Vigo (Galician upwelling ecosystem) in August 2013 (CHAOS cruises). During these cruises two 25-hour series of turbulent microstructure and currents observations were carried out during spring and neap tides. Turbulent kinetic energy dissipation at the interface between upwelled and surface waters was enhanced by two orders of magnitude during the ebbs, as the result of the interplay of the bi-directional upwelling circulation and the tidal currents shear (Fernández-Castro et al., 2018). This mechanism could have important implications for the functioning of biological processes, as it can act as a pathway for nutrient supply from upwelled nutrient-rich deep waters into the sunlit surface waters. In fact, diffusive nitrate fluxes due to the enhanced dissipation observed during CHAOS-springs, could be responsible for about half of the phytoplankton primary production estimated in this system during periods of upwelling relaxation-stratification (Villamaña et al., 2017). A new figure included in the supplementary material (Figure A2-C) indicates the number of turbulent profiles deployed at each station for each region. The duration of the microstructure turbulent profiler operation is indicated in Table 1 and the text has been modified to:

Original: Page 4 Line 28-29 Measurements of dissipation rates of turbulent kinetic energy ($\varepsilon$) were conducted at 3-10 profiles for each station to the bottom, or 300 m over deep waters.

Change:Page 4 Line 26- Page 5 Line 5 Measurements of dissipation rates of turbulent kinetic energy ($\varepsilon$) were conducted to the bottom, or to 137-323 m over deep waters (see Table 1). The microstructure turbulence profiles used for computing nitrate fluxes at each station were always deployed successively. Sets include 2-11 in the tropical and subtropical Atlantic (operation time 37 $\pm$ 18 min), 6-7 in the NW Mediterranean (76 $\pm$ 22 min) and 3-402 in the Galician coastal upwelling (65 $\pm$ 246 min) (Figure A2-C). Bursts of turbulence can induce episodic inputs of nutrient supply which can be easily missed in sets of low number of profiles. In coastal regions where short-term variability of mixing processes is expected to be higher, our dataset includes two highfrequency samplings carried out in the outer part of Ría de Vigo (Galician upwelling ecosystem) in August 2013 (CHAOS cruises). During these cruises two 25-hour series of turbulent microstructure and currents observations were carried out during spring and neap tides. Turbulent kinetic energy dissipation at the interface between upwelled and surface waters was enhanced by two orders of magnitude during the ebbs, as the result of the interplay of the bi-directional upwelling circulation and the tidal currents shear (Fernández-Castro et al., 2018).

4) PICOPHYTOPLANKTON BIOMASS The authors report the estimates of picophytoplankton biomass and ratios of biomass, but I was unable to find how the authors convert from cell abundance to carbon per unit volume. This is quite important, as there are several group-specific carbon conversion factors in the literature fand for the larger eukaryotic cells it is likely a biovolume conversion factor may provide a better estimate, as the size range within this subgroup can be significant.

We thank the reviewer for pointing out that this information was missing. In order to estimate biovolume (BV), we used an empirical calibration between Size SCatter (SSC) and cell diameter (Calvo-Díaz and Morán, 2006), assuming spherical shape for all groups. The following volume-to-carbon conversion factors were used for picoautotrophic groups: 230*fg C*BV for Synechococcus, 240*fg C*BV for Prochlorococcus and 237*fg C*BV for picoeukaryotes (Worden et al., 2004). For bacteria BV was converted into carbon biomass by using the allometric relationship: 108.8*fg C*BV0.898 (Gundersen et al., 2002).

Original: Page 6 Line 19-24 Autotrophic cells were separated into two groups of cyanobacteria (Synechococcus and Prochlorococcus) and one group of small picoeukaryotes, based on their fluorescence and light scatter signals (SSC), as explained in Calvo-Díaz et al. (2006). Two groups of heterotrophic prokaryotes (LNA and HNA) were distinguished based on their relative green fluorescence, which was used as a proxy for nucleic acid content (Gasol and del Giorgio, 2000; Bouvier et al., 2007).

Change: Page 7 Line 4-15 Autotrophic cells were separated into two groups of cyanobacteria (Synechococcus and Prochlorococcus) and one group of small pi-coeukaryotes, based on their fluorescence and light scatter signals (SSC), as explained in Calvo-Díaz et al. (2006). Two groups of heterotrophic prokaryotes (LNA and HNA) were distinguished based on their relative green fluorescence, which was used as a proxy for nucleic acid content (Gasol and del Giorgio, 2000; Bouvier et al., 2007). In order to estimate biovolume (BV), we used an empirical calibration between Size SCatter (SSC) and cell diameter (Calvo-Díaz et al., 2006), assuming spherical shape for all groups. The following volume-to-carbon conversion factors were used for picoauto-totrophic groups: 230 fg C for Synechococcus, 240 fg C for Prochlorococcus and 237 fg C for picoeukaryotes (Worden et al., 2004). For bacteria BV was converted into carbon biomass by using the allometric relationship: 108.8 fg C*BV0.898 (Gundersen et al., 2002). More details about the processing and analysis of flow cytometry samples are provided in Calvo-Díaz et al. (2006, TRYNITROP), Gomes et al. (2015, FAMOSO), Villamaña et al. (2017, CHAOS) and Moreira-Coello et al. (2017, NICANOR). Abundance data obtained at different depths for each station were combined to compute depth-integrated biomass for the photic layer.

5) EXTRAPOLATION TO THE GLOBAL OCEAN I was surprised to see that over half of the stations used were from coastal embayments. One could argue from many points of view that these regions may not be representative of the open-ocean eu-trophic environments (likely different taxonomic diversity within these gross cytometric groupings, potential supply of nutrients from terrigenous sources, different light envi-ronment caused by attenuation by CDOM and sediments, need to correct for advective flux). Perhaps the authors have supporting literature/data that would help convince the reader that these embayments broadly reflect the environmental conditions of offshore stations, but even with such supporting information they should also highlight the need for data from open-ocean meso- and eutrophic environments that would help further resolve the global relationship between mixing and picoplankton community structure.

We agree with this reviewer that a larger data set including a wider range of conditions will be desirable. However, please note that the Galician Rías, despite being in general longer and narrower than many open bays in upwelling areas (e.g., Monterey Bay in California, False Bay in South Africa, Antofagasta Bay in Chile, and Todos Santos Bay in Mexico), they resemble them in that its primary hydrographic and circulation features are determined by the extension of wind-driven flow on the external continental shelf throughout the bay (Alvarez-Salgado et al., 2010). Fertilization in the Rías occurs essentially by coastal upwelling, being fresh and rain water inputs residual (2

Original Page 5 Line 21-25 Most stations carried out in the Galician coastal upwelling were conducted inside three different Rías (Ría de Vigo, Ría de Pontevedra and Ría de A Coruña). The Rías are coastal embayments affected by seasonal wind-driven coastal upwelling of cold, nutrient-rich North Atlantic Central water (Wooster et al., 1976; Fraga, 1981; Álvarez-Salgado et al., 1993). The total nitrate supply in the Galician Rías was computed as the sum of nitrate vertical diffusion plus nitrate vertical advection due to coastal upwelling.

Modify Page 6 Line 5 - 12 Most stations carried out in the Galician coastal upwelling were conducted inside three different Rías (Ría de Vigo, Ría de Pontevedra and Ría de A Coruña). The Rías are coastal embayments affected by seasonal wind-driven coastal upwelling of cold, nutrient-rich North Atlantic Central water (Wooster et al., 1976; Fraga, 1981; Álvarez-Salgado et al., 1993). The Galician Rías, despite being in general longer and narrower than many open bays in upwelling areas, they resemble them in that its primary hydrographic and circulation features are determined by the extension of wind-driven flow on the external continental shelf throughout the bay (Alvarez-Salgado et al., 2010). Fertilization in the Rías occurs essentially by coastal upwelling, being fresh and rain water inputs residual (2as the sum of nitrate vertical diffusion plus nitrate vertical advection due to coastal upwelling.

The dataset is largely confined to a specific geographic region (40N-30S, covering a limited number of biochemical provinces in the Atlantic Basin), yet the authors use rela-

tionships from this study to predict future changes in picoplankton community structure across the entire globe. Would the authors consider limiting their predictions of future community structure to the geographic regions/ latitudinal gradients that are used to develop the predictive models? Given that the global ocean covers a variety of biogeochemical regimes, some of which may not be limited by nitrate, restricting the geographical scope of the future predictions may be advisable, even though the overall patterns tend to broadly resemble those from other global studies.

Following the comments by both reviewers this section has been deleted from the manuscript.

6) USE OF MOREL MODEL TO ESTIMATE DEPTH OF PHOTIC LAYER Is the light attenuation observed in the Galician coastal stations largely a result of phytoplankton or other optically-active substances? I mention this because the model of Morel used to estimate euphotic depth is restricted to Case-1 (open ocean) waters where light attenuation is dominated by phytoplankton.

As mentioned above primary hydrographic and circulation features in the Galician Rías are determined by the extension of wind-driven flow on the external continental shelf throughout the bay. Note that light attenuation is exclusively used in the manuscript to get an estimate of the base of the euphotic layer, to be used as the lower limit to compute depth-integrated biomass. Using water type 2 equations would result in a shallower limit for the euphotic layer, which could miss phytoplankton biomass sometimes located deeper that this limit in this system (see Figure 1 in Cermeño et al., (2016)). In our study the Morel equation was only used to compute light attenuation coefficients during the NICANOR, ASIMUTH and CHAOS cruises, which sampled stations in the outer part of the Rias. This information is now clarified in the methods section. A new figure (Figure A2-D) has been also included in the supplementary material to compare the base of the euphotic zone, as derived from PAR data and the Morel equation, by using data collected during the HERCULES cruises.

Original Pag 4 Line 20-24 For those cruises where PAR profiles were not available (ASIMUTH, CHAOS and NICANOR), the depth of the photic layer was calculated by considering light attenuation coefficients derived from surface chlorophyll-a data estimated from the space, following the algorithms proposed by Morel et al. (2007) (http://globcolour.info).

Modified Pag 4 Line 17-22 For those cruises where PAR profiles were not available (ASIMUTH, CHAOS and NICANOR), which sampled stations in the outer part of the Rias, the depth of the photic layer was calculated by considering light attenuation coefficients derived from surface chlorophyll-a data, following the algorithms proposed by Morel et al. (2007) for Case-1 waters (log10Zeu=$1.524 - 0.460$[Chl]surf $- 0.00051$[Chl]2surf$+ 0.0282$[Chl]3surf). A comparison of the estimation of the base of the euphotic zone by using the Morel et al equation and the data collected by a radiometer during the HERCULES cruise is shown in Figure A2-D.

---

## Author Comment (AC2) · 14 Sep 2018

Anonymous Referee 1 Referee comments shown in black, Author replies shown in blue, Changes to manuscript in red

Overall comments

The manuscript investigates the relationship between nutrient supply estimated by measurements of turbulent mixing and nitrate gradients and the community structure of picophytoplankton (including both autotrophic and heterotrophic groups). The take-home message of the work is that studies that use ambient nutrient concentrations as

a proxy of nutrient availability could be misleading as in many regions of the ocean the supply of nutrients by turbulent diffusion is not registered in bottle samples of nutrients. This is an important message that needs to be communicated to the wider marine science community.

The manuscript is well written and provides a nice overview of the ecological literature of marine picophytoplankton. The dataset of turbulent mixing, nitrate concentrations and picoplankton community structure is novel and covers a variety of hydrographic and trophic regimes.

I have a few questions and comments concerning both the estimation of nutrient supply using combined MSS and nutrient profiles and the choice of sampling stations used in the analysis which I hope the authors may be able to address.

Specific comments

1) NUTRIENT FLUX ESTIMATES

Although the authors correctly point out that concomitant datasets of turbulent mixing and picophytoplankton community structure are rare, this may be due in part to the lack of high-quality estimates of nutrient flux from microstructure profile measurements in the surface ocean with coincident depth-resolved nutrient profiles (required to obtain a robust estimate of the nutrient gradient near the nutricline). The vertical resolution of the nutrient profiles within the dataset is unclear (only a range between 3 and 9 depths is provided, but it could be tricky to use 3/4 depths to provide a good estimate of the nutrient gradient ). Could the extent of the density gradient be over/underestimated in cases where the depth resolution is low and by how much? It would be good to have a frequency histogram for the dataset showing the number of depths per profile so the reader is aware of the vertical resolution of nutrient concentrations across the dataset.

We agree with the reviewer that the lower vertical resolution of nitrate concentration in comparison to turbulent data incorporates some uncertainty, which is very hard to

evaluate, in the calculation of nitrate diffusive fluxes. Lower vertical resolution (5±2 sampling depths) was used in the shallower stations sampled in the Galician coastal upwelling, whereas in the deeper NW Mediterranean and the tropical and subtropical regions we used 7±1 and 11±2 depths, respectively. The text has been modified and a new figure (Figure A2.A) has been included in the supplementary material, to represent the frequency histogram of sampling depths used to compute the nitrate vertical gradient in each region The maximum sampling depth where the microstructure turbulence profiler was deployed is now indicated in Table 1.

Original Page 5 – Line 11-13 Samples for the determination of nitrate (NO3) + nitrite (NO2 ) were collected from 3 to 9 depths in rinsed polyethylene tubes and stored frozen at -20 °C until analysis on land, according to standard methods using the automated colorimetric technique (Grasshoff et al., 2007).

Modified to Page 5 – Line 21-26 Samples for the determination of nitrate (NO3) + nitrite (NO2 ) were collected, on average, from 5±2 (Galician coastal upwelling), 7±1 (NW Mediterranean) and 11±2 (tropical and subtropical Atlantic Ocean) depths in rinsed polyethylene tubes and stored frozen at -20 °C until analysis on land, according to standard methods using the automated colorimetric technique (Grasshoff et al., 2007). The frequency histogram of sampling depths collected for the determination of nitrate concentration in each region is indicated in Figure A2-A in the Supplementary material, whereas the maximum sampling depth where the microstructure turbulence profiler was deployed is indicated in Table 1.

2) DENSITY AND NITRATE RELATIONSHIPS The authors also mention that for some of the stations nutrient data was not available, and instead of nutrient bottle data, a relationship between density and nitrate was used. Again it is not clear how robust the relationship between nitrate concentration and density was for the relevant stations. Could the authors provide supplementary plots of the nutrient versus density relationship that was used to estimate nitrate gradients, similar to that of Williams et al. (2013a GRL 40:5467-5472; 2013b Limnol. Oceanogr.: Fluids and Envs 3:61-73)?

Only in one station sampled in the Galician coastal upwelling ecosystem during the NICANOR project observations of nutrient concentration were not available. Instead nitrate concentration was computed from a nitrate-density (sigma-t) relationship built by using all samples (n=52) collected during the NICANOR sampling period. A new figure (Figure A2-B) has been included in the supplementary material to show this relationship. The relationship showed a linear behavior (NO$-3$ = 9.7788 *sigma-t $-$ 256.38; r =0.930; p < 0.001) for density ranging between 26.1 and 27.1 kg m$-3$.

Original: Page 5 – Line 13-14 In those stations carried out during the NICANOR cruises, where nitrate concentrations were not available, they were obtained by using nitrate-density relationships (Moreira-Coello et al., 2017)

Change: Page 5 – Line 26-29 In one station carried out during the NICANOR cruise, where nitrate concentrations were not available, these were obtained by using a nitrate-density relationship built by using all samples (n=52) collected during the NICANOR sampling period. The relationship showed a linear behavior (NO3 = 9.7788 *$\sigma$t $-$ 256.38; Adj-r2 =0.87; p < 0.001) for density ranging between 26.1 and 27.1 kg m$-3$ (Figure A2-B).

3) EPISODIC NATURE OF MIXING Mixing events in some regions can be episodic, yet short-term vertical pulses of nutrients can trigger significant shifts in community structure. In some oceanic regions tidal mixing can also be important. The authors mention that 3-10 profiles were taken, but it would also be helpful to know the time interval over which these profiles were made and how this varied between the three study regions (again a frequency histogram documenting this would be helpful). Could it be that for some regions the flux of nutrients could be significantly underestimated given that such short-term events may not be captured in MSS profile data? Given the general audience of the journal, both the strengths and shortcomings of this method of estimating nutrient supply should be provided.

We completely agree with the reviewer that episodic bursts of turbulence can induce

episodic inputs of nutrient supply which can be easily missed in sets of 2-10 profiles. The microstructure turbulence profiles used for computing nitrate fluxes at each station were always deployed successively. Sets include 2-11 in the tropical and subtropical Atlantic ($37 \pm 18$ min), 6-7 in the NW Mediterranean ($76 \pm 22$ min) and 3-402 in the Galician coastal upwelling ($65 \pm 246$ min). Our dataset includes two high-frequency samplings carried out in the outer part of Ría de Vigo (Galician upwelling ecosystem) in August 2013 (CHAOS cruises). During these cruises two 25-hour series of turbulent microstructure and currents observations were carried out during spring and neap tides. Turbulent kinetic energy dissipation at the interface between upwelled and surface waters was enhanced by two orders of magnitude during the ebbs, as the result of the interplay of the bi-directional upwelling circulation and the tidal currents shear (Fernández-Castro et al., 2018). This mechanism could have important implications for the functioning of biological processes, as it can act as a pathway for nutrient supply from upwelled nutrient-rich deep waters into the sunlit surface waters. In fact, diffusive nitrate fluxes due to the enhanced dissipation observed during CHAOS-springs, could be responsible for about half of the phytoplankton primary production estimated in this system during periods of upwelling relaxation-stratification (Villamaña et al., 2017). A new figure included in the supplementary material (Figure A2-C) indicates the number of turbulent profiles deployed at each station for each region. The duration of the microstructure turbulent profiler operation is indicated in Table 1 and the text has been modified to:

Original: Page 4 Line 28-29 Measurements of dissipation rates of turbulent kinetic energy ($\varepsilon$) were conducted at 3-10 profiles for each station to the bottom, or 300 m over deep waters.

Change:Page 4 Line 26- Page 5 Line 5 Measurements of dissipation rates of turbulent kinetic energy ($\varepsilon$) were conducted to the bottom, or to 137-323 m over deep waters (see Table 1). The microstructure turbulence profiles used for computing nitrate fluxes at each station were always deployed successively. Sets include 2-11 in the tropical

and subtropical Atlantic (operation time 37 $\pm$ 18 min), 6-7 in the NW Mediterranean (76 $\pm$ 22 min) and 3-402 in the Galician coastal upwelling (65 $\pm$ 246 min) (Figure A2-C).

Bursts of turbulence can induce episodic inputs of nutrient supply which can be easily missed in sets of low number of profiles. In coastal regions where short-term variability of mixing processes is expected to be higher, our dataset includes two high-frequency samplings carried out in the outer part of Ría de Vigo (Galician upwelling ecosystem) in August 2013 (CHAOS cruises). During these cruises two 25-hour series of turbulent microstructure and currents observations were carried out during spring and neap tides. Turbulent kinetic energy dissipation at the interface between upwelled and surface waters was enhanced by two orders of magnitude during the ebbs, as the result of the interplay of the bi-directional upwelling circulation and the tidal currents shear (Fernández-Castro et al., 2018).

4) PICOPHYTOPLANKTON BIOMASS The authors report the estimates of picophytoplankton biomass and ratios of biomass, but I was unable to find how the authors convert from cell abundance to carbon per unit volume. This is quite important, as there are several group-specific carbon conversion factors in the literature fand for the larger eukaryotic cells it is likely a biovolume conversion factor may provide a better estimate, as the size range within this subgroup can be significant.

We thank the reviewer for pointing out that this information was missing. In order to estimate biovolume (BV), we used an empirical calibration between Size SCatter (SSC) and cell diameter (Calvo-Díaz and Morán, 2006), assuming spherical shape for all groups. The following volume-to-carbon conversion factors were used for picoautotrophic groups: 230*fg C*BV for Synechococcus, 240*fg C*BV for Prochlorococcus and 237*fg C*BV for picoeukaryotes (Worden et al., 2004). For bacteria BV was converted into carbon biomass by using the allometric relationship: 108.8*fg C*BV0.898 (Gundersen et al., 2002).

Original: Page 6 Line 19-24 Autotrophic cells were separated into two groups of

cyanobacteria (Synechococcus and Prochlorococcus) and one group of small pi-coeukaryotes, based on their fluorescence and light scatter signals (SSC), as explained in Calvo-Díaz et al. (2006). Two groups of heterotrophic prokaryotes (LNA and HNA) were distinguished based on their relative green fluorescence, which was used as a proxy for nucleic acid content (Gasol and del Giorgio, 2000; Bouvier et al., 2007).

Change: Page 7 Line 4-15 Autotrophic cells were separated into two groups of cyanobacteria (Synechococcus and Prochlorococcus) and one group of small pi-coeukaryotes, based on their fluorescence and light scatter signals (SSC), as explained in Calvo-Díaz et al. (2006). Two groups of heterotrophic prokaryotes (LNA and HNA) were distinguished based on their relative green fluorescence, which was used as a proxy for nucleic acid content (Gasol and del Giorgio, 2000; Bouvier et al., 2007). In order to estimate biovolume (BV), we used an empirical calibration between Size SCatter (SSC) and cell diameter (Calvo-Díaz et al., 2006), assuming spherical shape for all groups. The following volume-to-carbon conversion factors were used for picoau-totrophic groups: 230 fg C for Synechococcus, 240 fg C for Prochlorococcus and 237 fg C for picoeukaryotes (Worden et al., 2004). For bacteria BV was converted into carbon biomass by using the allometric relationship: 108.8 fg C*BV0.898 (Gundersen et al., 2002). More details about the processing and analysis of flow cytometry samples are provided in Calvo-Díaz et al. (2006, TRYNITROP), Gomes et al. (2015, FAMOSO), Villamaña et al. (2017, CHAOS) and Moreira-Coello et al. (2017, NICANOR). Abundance data obtained at different depths for each station were combined to compute depth-integrated biomass for the photic layer.

5) EXTRAPOLATION TO THE GLOBAL OCEAN I was surprised to see that over half of the stations used were from coastal embayments. One could argue from many points of view that these regions may not be representative of the open-ocean eu-trophic environments (likely different taxonomic diversity within these gross cytometric groupings, potential supply of nutrients from terrigenous sources, different light environment caused by attenuation by CDOM and sediments, need to correct for advective

flux). Perhaps the authors have supporting literature/data that would help convince the reader that these embayments broadly reflect the environmental conditions of offshore stations, but even with such supporting information they should also highlight the need for data from open-ocean meso- and eutrophic environments that would help further resolve the global relationship between mixing and picoplankton community structure.

We agree with this reviewer that a larger data set including a wider range of conditions will be desirable. However, please note that the Galician Rías, despite being in general longer and narrower than many open bays in upwelling areas (e.g., Monterey Bay in California, False Bay in South Africa, Antofagasta Bay in Chile, and Todos Santos Bay in Mexico), they resemble them in that its primary hydrographic and circulation features are determined by the extension of wind-driven flow on the external continental shelf throughout the bay (Alvarez-Salgado et al., 2010). Fertilization in the Rías occurs essentially by coastal upwelling, being fresh and rain water inputs residual (2

Original Page 5 Line 21-25 Most stations carried out in the Galician coastal upwelling were conducted inside three different Rías (Ría de Vigo, Ría de Pontevedra and Ría de A Coruña). The Rías are coastal embayments affected by seasonal wind-driven coastal upwelling of cold, nutrient-rich North Atlantic Central water (Wooster et al., 1976; Fraga, 1981; Álvarez-Salgado et al., 1993). The total nitrate supply in the Galician Rías was computed as the sum of nitrate vertical diffusion plus nitrate vertical advection due to coastal upwelling.

Modify Page 6 Line 5 - 12

Most stations carried out in the Galician coastal upwelling were conducted inside three different Rías (Ría de Vigo, Ría de Pontevedra and Ría de A Coruña). The Rías are coastal embayments affected by seasonal wind-driven coastal upwelling of cold, nutrient-rich North Atlantic Central water (Wooster et al., 1976; Fraga, 1981; Álvarez-Salgado et al., 1993). The Galician Rías, despite being in general longer and narrower than many open bays in upwelling areas, they resemble them in that its primary hydrographic and circulation features are determined by the extension of wind-driven flow on the external continental shelf throughout the bay (Alvarez-Salgado et al., 2010). Fertilization in the Rías occurs essentially by coastal upwelling, being fresh and rain water inputs residual (2

The dataset is largely confined to a specific geographic region (40N-30S, covering a limited number of biochemical provinces in the Atlantic Basin), yet the authors use relationships from this study to predict future changes in picoplankton community structure across the entire globe. Would the authors consider limiting their predictions of future community structure to the geographic regions/ latitudinal gradients that are used to develop the predictive models? Given that the global ocean covers a variety of biogeochemical regimes, some of which may not be limited by nitrate, restricting the geographical scope of the future predictions may be advisable, even though the overall patterns tend to broadly resemble those from other global studies.

Following the comments by both reviewers this section has been deleted from the manuscript.

6) USE OF MOREL MODEL TO ESTIMATE DEPTH OF PHOTIC LAYER Is the light attenuation observed in the Galician coastal stations largely a result of phytoplankton or other optically-active substances? I mention this because the model of Morel used to estimate euphotic depth is restricted to Case-1 (open ocean) waters where light attenuation is dominated by phytoplankton.

As mentioned above primary hydrographic and circulation features in the Galician Rías are determined by the extension of wind-driven flow on the external continental shelf throughout the bay. Note that light attenuation is exclusively used in the manuscript to get an estimate of the base of the euphotic layer, to be used as the lower limit to compute depth-integrated biomass. Using water type 2 equations would result in a shallower limit for the euphotic layer, which could miss phytoplankton biomass sometimes located deeper that this limit in this system (see Figure 1 in Cermeño et al., (2016)). In

our study the Morel equation was only used to compute light attenuation coefficients during the NICANOR, ASIMUTH and CHAOS cruises, which sampled stations in the outer part of the Rias. This information is now clarified in the methods section. A new figure (Figure A2-D) has been also included in the supplementary material to compare the base of the euphotic zone, as derived from PAR data and the Morel equation, by using data collected during the HERCULES cruises.

Original Pag 4 Line 20-24 For those cruises where PAR profiles were not available (ASIMUTH, CHAOS and NICANOR), the depth of the photic layer was calculated by considering light attenuation coefficients derived from surface chlorophyll-a data estimated from the space, following the algorithms proposed by Morel et al. (2007) (http://globcolour.info).

Modified Pag 4 Line 17-22 For those cruises where PAR profiles were not available (ASIMUTH, CHAOS and NICANOR), which sampled stations in the outer part of the Rias, the depth of the photic layer was calculated by considering light attenuation coefficients derived from surface chlorophyll-a data, following the algorithms proposed by Morel et al. (2007) for Case-1 waters ($\log_{10}Z_{eu}=1.524 - 0.460[Chl]_{surf} - 0.00051[Chl]2_{surf} + 0.0282[Chl]3_{surf}$). A comparison of the estimation of the base of the euphotic zone by using the Morel et al equation and the data collected by a radiometer during the HERCULES cruise is shown in Figure A2-D.
* * *
**Table 1.** Details of the data included in this study. Domain referred to the tropical and subtropical Atlantic ocean (T), the Northwestern Mediterranean Sea (M), and the Galician coastal upwelling (G). N indicates the number of stations sampled at each cruise. Duration (mean ± standard deviation) is the time duration in minutes of the turbulence profiler deployment in each station. Duration (mean ± standard deviation, in minutes) is the time used for the microstructure turbulence operation at each station). Depth (mean ± standard deviation, in meters) is the maximum depth reached by the microstructure profiler.

| Domain | Region | N | Cruise | Vessel | Date | Duration | Depth |
|--------|--------|---|--------|--------|------|----------|-------|
| T | NE Atlantic | 8 | CARPOS | Hespérides | 14/10/06- 22/11/06 | 57 ± 24 | 137 ± 15 |
| T | Atlantic | 18 | TRYNITROP | Hespérides | 14/04/08 - 02/05/08 | 45 ± 12 | 219 ± 19 |
| M | Liguro-Provençal Basin | 6 | FAMOSO I | Sarmiento de Gamboa | 14/3/09 - 22/3/09 | 66 ± 5 | 259 ± 38 |
| M | Liguro-Provençal Basin | 10 | FAMOSO II | Sarmiento de Gamboa | 30/4/09 - 13/05/09 | 94 ± 4 | 273 ± 2 |
| M | Liguro-Provençal Basin | 3 | FAMOSO III | Sarmiento de Gamboa | 16/09/09 - 20/09/09 | 133 ± 3 | 323 ± 24 |
| G | Ría de A Coruña | 1 | HERCULES I | Lura | 07/06/10 | 20 ± 4 | 35 ± 2 |
| G | Ría de A Coruña | 5 | HERCULES II | Lura | 28/09/11 - 29/09/11 | 11 ± 8 | 33 ± 26 |
| G | Ría de A Coruña | 13 | HERCULES III | Lura | 16/07/12 - 20/07/12 | 8 ± 5 | 41 ± 29 |
| G | Ría de Vigo | 9 | DISTRAL | Mytilus | 14/02/12 - 06/11/12 | 110 ± 76 | 38 ± 1 |
| G | Ría de Vigo | 2 | CHAOS | Mytilus | 20/08/13 - 27/08/13 | 1515 ± 6 | 41 ± 29 |
| G | Ría de A Coruña | 12 | NICANOR | Lura | 27/02/14 - 17/12/15 | 33 ± 5 | 62 ± 3 |
| G | Rías de Vigo & Pontevedra | 10 | ASIMUTH | Ramón Margalef | 17/06/13 - 21/06/13 | 10 ± 4 | 28 ± 10 |

**Fig. 1.** Table 1

[Figure]

**Figure A2.** A) Frequency histograms of the number of nutrient where samples for nitrate concentration were collected at each station and domain: tropical and subtropical Atlantic ocean (red), the Northwestern Mediterranean (green) and Galician coastal upwelling (blue). B) Pair scatter plot representing the relationship between nitrate concentration and density built by using all samples collected during the NICANOR sampling period. C) Frequency histogram of the number of turbulence profiles deployed at each station and domain. D) Pair scatter plot representing the relationship between the euphotic zone depth ($Z_{eu}$) computed using the Morel et al. (2007) equation and the data collected by a radiometer during the HERCULES cruise measured used a radiometer and predicted using the relationship with surface chlorophyll Morel et al. (2007), the solid line represents 1:1 relationship.

**Fig. 2.** Figure A2

[Figure]

[Figure]

**Figure A3.** Vertical distribution of temperature (Temp), nitrate (NO₃) and picoplankton biomass of autotrophic (Phyto) and heterotrophic (Bacteria) groups for each domain: tropical and subtropical Atlantic ocean (T), the Northwestern Mediterranean (M), and Galician coastal upwelling (G). Points represent raw data and the solid line the locally weighted scatterplot smoothing (LOESS). Dashed lines indicate 95% confidence intervals. Dot and line color intensity indicates the number of overlapping observations.

**Fig. 3.** Figure A3

---

## Author Comment (AC4) · 15 Sep 2018

Referee comments shown in black, Author replies shown in blue, Changes to manuscript in red

Overall comments

The manuscript by Otero-Ferrer et al reports the relationship between nitrate supply and temperature in the structure of picoplankton groups determined by flow cytometer. The manuscript is really well written, and the literature seems extensively covered. The strength of this work resides in the use of nitrate diffuse flux as a proxy of nutrient

availability and depth integrate biomass for the different picoplankton groups to predict the niche of the groups analyzed. Commonly, these measurements are treated as discrete rather than continuous variables. The data and results presented by Otero-Ferrer et al have lot of potential and I am confident that the community will benefit from its publication. However, I believe there are very few points that can be amended in a way to improve clarity of the main message of this work.

1) Plots with the vertical distribution of the main variables considered (nitrate, cell abundance of picoplankton groups and temperature) could be provided and would certainly help the reader to have a better assessment of the conditions in the sampled stations.

A new figure (Figure A3) has been include in the supplementary material to show the vertical distribution of temperature, nitrate concentration and picoplankton biomass of autotrophic and heterotrophic groups. References to the new figure have been included in the manuscript:

Original:

Page 13 Line 7-10

It is also important to note that surface abundance of picoplankton subgroups reported in our study, which are consistent with previous observations Zubkov et al. (2000); Frojan et al. (2014); Teira et al. (2015), did show higher surface abundance of picoeukaryotes in the Galicia coastal upwelling and the NW Mediterranean compared to the tropical and subtropical Atlantic (Table 2).

Page 9 Lines 8-9

Finally, HNA (55%) and LNA (21%) prokaryotes dominated in the Galician coastal upwelling system, followed by picoeukaryotes (11%), *Synechococcus* (6%) and *Prochlorococcus* (1%).

Change:

Page 13 Lines 8-12

It is also important to note that surface abundance of picoplankton subgroups reported in our study, which are consistent with previous observations Zubkov et al. (2000); Frojan et al. (2014); Teira et al. (2015), did show higher surface abundance of picoeukaryotes in the Galicia coastal upwelling and the NW Mediterranean compared to the tropical and subtropical Atlantic (see Table 2 and Figure A3).

Page 9 Lines 10-14

Finally, HNA (55%) and LNA (21%) prokaryotes dominated in the Galician coastal upwelling system, followed by picoeukaryotes (11%), *Synechococcus* (6%) and *Prochlorococcus* (1%).

Vertical distributions of temperature, nitrate concentration and the biomass of autotrophic and heterotrophic picoplankton groups are shown in Figure A3.

2) Which values of cell-to-carbon conversion factors were used to transform abundance into biomass of the different groups of picoplankton analyzed?

We thank the reviewer for pointing out that this information was missing. In order to estimate biovolume (BV), we used an empirical calibration between Size SCatter (SSC) and cell diameter (Calvo-Díaz and Morán, 2006), assuming spherical shape for all groups. The following volume-to-carbon conversion factors were used for picoautotrophic groups: 230*fg C*BV for *Synechococcus*, 240*fg C*BV for *Prochlorococcus* and 237*fg C*BV for picoeukaryotes (Worden et al., 2004). For bacteria BV was converted into carbon biomass by using the allometric relationship: 108.8*fg C*BV0.898 (Gundersen et al., 2002).

Original: Page 6 Line 19-24

Autotrophic cells were separated into two groups of cyanobacteria (*Synechococcus* and *Prochlorococcus*) and one group of small picoeukaryotes, based on their fluorescence and light scatter signals (SSC), as explained in Calvo-Díaz et al. (2006). Two

groups of heterotrophic prokaryotes (LNA and HNA) were distinguished based on their relative green fluorescence, which was used as a proxy for nucleic acid content (Gasol and del Giorgio, 2000; Bouvier et al., 2007).

Change: Page 7 Line 4-15

Autotrophic cells were separated into two groups of cyanobacteria (*Synechococcus* and *Prochlorococcus*) and one group of small picoeukaryotes, based on their fluorescence and light scatter signals (SSC), as explained in Calvo-Díaz et al. (2006). Two groups of heterotrophic prokaryotes (LNA and HNA) were distinguished based on their relative green fluorescence, which was used as a proxy for nucleic acid content (Gasol and del Giorgio, 2000; Bouvier et al., 2007).

In order to estimate biovolume (BV), we used an empirical calibration between Size SCatter (SSC) and cell diameter (Calvo-Díaz et al., 2006), assuming spherical shape for all groups. The following volume-to-carbon conversion factors were used for picoautotrophic groups: 230 fg C for *Synechococcus*, 240 fg C for *Prochlorococcus* and 237 fg C for picoeukaryotes (Worden et al., 2004). For bacteria BV was converted into carbon biomass by using the allometric relationship: 108.8 fg C*BV0.898 (Gundersen et al., 2002). More details about the processing and analysis of flow cytometry samples are provided in Calvo-Díaz et al. (2006, TRYNITROP), Gomes et al. (2015, FAMOSO), Villamaña et al. (2017, CHAOS) and Moreira-Coello et al. (2017, NICANOR). Abundance data obtained at different depths for each station were combined to compute depth-integrated biomass for the photic layer.

3) The depth integrated biomass of picoeuks was not linearly correlated with temperature, PAR nor nitrate flux when the authors used the simple linear model. However, with the additive model, picoeuks showed a negative trend with temperature and unimodal distribution with nitrate. Could the authors elaborate a bit more in the discussion about this contradiction between methods?

There is no contradiction between the results from the Generalized Linear and Generalized Additive Models. Generalized Additive Models are a form of regression analysis in which observational data are modeled by a function which is a nonlinear combination of the model parameters, and depends on one or more independent variables. By contrast, Generalized Linear Models generally include a nonlinear relationship between response and predictors, but the link-transformed mean response is linear in the parameters. If parameters have a linear behavior, partial effects will show a linear relationship.

This was indicated in the previous draft in the following sections:

Page 6, Lines 26 - Page 7 Line 9: "A Generalized Additive Model (GAM) approach was used to predict depth-integrated biomass of each picoplankton subgroup, the contribution of LNA prokaryotes to heterotrophic picoplankton, the cyanobacteria to picoeukaryotes ratio, and the autotrophic to heterotrophic ratio based on observations and estimates of three environmental factors: sea surface temperature (SST), daily surface PAR, and the transport of nitrate into the euphotic zone ($NO_3Flux$), including both diffusive and advective processes. GAMs assume that the effect of each predictor on the response variable can be described by smoothed functions whose effects are additive. Due to the large number of zero observations, data overdispersion, and the need for a single parsimonious model to make predictions for a large number of groups, we assumed that the depth-integrated biomass of each picoplankton subgroup, relative contribution values and biomass ratios all followed negative binomial distributions. Those niche descriptors that did not follow normal distributions were log transformed. The complete model structure for the biomass of each picoplankton subgroup was:

$$y_j = I + s(SST) + s(PAR) + s(log(NO_3Flux)) + Error$$

where y represents the depth-integrated biomass for each picoplankton subgroup j, and s a cubic regression spline used for fitting the observations to the model (Wood, 2006). Generalized models include a function linking the mean value of yj and the predictors. For those response variables that followed a negative binomial distribution the used

link function was the natural logarithm. The LNA contribution to total heterotrophic prokaryotes was adjusted using a gaussian distribution and an identify link (Wood et al., 2016)."

Page 9, Lines 24-25:

"In order to exclude cross-correlation between the three environmental factors and consider the possibility of non-linear relationships, we subsequently fitted the data to Generalized Additive Models (Figure 4 and Table 3).

4) The dataset of this manuscript was originated from coastal waters rather than oceanic, from two oceanic regions (Atlantic and Med Sea) and it is confined to a narrow latitude range. Thus It does not support extrapolations to worldwide oceans. I recommend the authors to be more caution and remove figure A2 and the lines 23 to 27 of the last paragraph.

Attending to the referee advise we remove this part of the text.

Specific comments INTRO Line 26 – missing a space between the word communities and the reference.

Done

MM Section 2, Line 5 – please keep one abbreviation for the Med sea to avoid confusion by writing the only once northwestern. No need to repeat every time since for the Atlantic the same was done.

Done

Section 2, line 14 – Diaz et al 2018 does not seem to be on bioxriv or any other repository, thus the info is not available. I would not cite unless the paper has been already released.

This citation has been deleted

Section 2.6, line 6 – add Generalized Linear Models before GLM abbreviation since it is the 1st time that appears.

Done

RESULTS Section 4.1, line 15, please add the average temperature value for picoeuks, especially because it seems very close to the one found for syn.

*Synechococcus* and picoeukaryotes niches overlap when temperature is considered. For this reason it does not seem necessary to report temperature value for these groups.

Original: Page 11 Line 13-15

*Synechococcus* and HNA prokaryotes prevailed mainly in cooler (below 20°C) marine environments characterized by intermediate and high levels of nitrate supply, and finally, the niche for picoeukaryotes was characterized by lower temperatures and high nitrate supply.

Change:Page 11 Line 13-15

*Synechococcus* and HNA prokaryotes prevailed mainly in cooler (below 20°C) marine environments characterized by intermediate and high levels of nitrate supply, and finally, the niche for picoeukaryotes was characterized by low temperatures and high nitrate supply.

DISCUSSION The figures and tables still can be cited in the discussion section. It facilitates a lot the follow up of the points discussed.

Cites referring key figures in the discussion have been included.

Bibliography

Calvo-Díaz, A. and Morán, X. A. G.: Seasonal dynamics of picoplankton in shelf waters of the southern Bay of Biscay, Aquat. Microb. Ecol., 42(2), 159–174,

doi:10.3354/ame042159, 2006.

Gundersen, K., Heldal, M., Norland, S., Purpie, D. A. and Knap, A. N.: Elemental C, N, and P cell content of individual bacteria collected at the Bermuda Atlantic Time-series Study (BATS) site, Limnol. Oceanogr., 47(5), 1525–1530 [online] Available from: http://cat.inist.fr/?aModele=afficheNcpsidt=13918565 (Accessed 18 June 2014), 2002.

Worden, A. Z., Nolan, J. K. and Palenik, B.: Assessing the dynamics and ecology of marine picophytoplankton: The importance of the eukaryotic component, Limnol. Oceanogr., 49(1), 168–179, 2004.
* * *
**Table 1.** Details of the data included in this study. Domain referred to the tropical and subtropical Atlantic ocean (T), the Northwestern Mediterranean Sea (M), and the Galician coastal upwelling (G). N indicates the number of stations sampled at each cruise. Duration (mean ± standard deviation) is the time duration in minutes of the turbulence profiler deployment in each station. Duration (mean ± standard deviation, in minutes) is the time used for the microstructure turbulence operation at each station). Depth (mean ± standard deviation, in meters) is the maximum depth reached by the microstructure profiler.

| Domain | Region | N | Cruise | Vessel | Date | Duration | Depth |
|--------|--------|---|--------|--------|------|----------|-------|
| T | NE Atlantic | 8 | CARPOS | Hespérides | 14/10/06- 22/11/06 | 57 ± 24 | 137 ± 15 |
| T | Atlantic | 18 | TRYNITROP | Hespérides | 14/04/08 - 02/05/08 | 45 ± 12 | 219 ± 19 |
| M | Liguro-Provençal Basin | 6 | FAMOSO I | Sarmiento de Gamboa | 14/3/09 - 22/3/09 | 66 ± 5 | 259 ± 38 |
| M | Liguro-Provençal Basin | 10 | FAMOSO II | Sarmiento de Gamboa | 30/4/09 - 13/05/09 | 94 ± 4 | 273 ± 2 |
| M | Liguro-Provençal Basin | 3 | FAMOSO III | Sarmiento de Gamboa | 16/09/09 - 20/09/09 | 133 ± 3 | 323 ± 24 |
| G | Ría de A Coruña | 1 | HERCULES I | Lura | 07/06/10 | 20 ± 4 | 35 ± 2 |
| G | Ría de A Coruña | 5 | HERCULES II | Lura | 28/09/11 - 29/09/11 | 11 ± 8 | 33 ± 26 |
| G | Ría de A Coruña | 13 | HERCULES III | Lura | 16/07/12 - 20/07/12 | 8 ± 5 | 41 ± 29 |
| G | Ría de Vigo | 9 | DISTRAL | Mytilus | 14/02/12 - 06/11/12 | 110 ± 76 | 38 ± 1 |
| G | Ría de Vigo | 2 | CHAOS | Mytilus | 20/08/13 - 27/08/13 | 1515 ± 6 | 41 ± 29 |
| G | Ría de A Coruña | 12 | NICANOR | Lura | 27/02/14 - 17/12/15 | 33 ± 5 | 62 ± 3 |
| G | Rías de Vigo & Pontevedra | 10 | ASIMUTH | Ramón Margalef | 17/06/13 - 21/06/13 | 10 ± 4 | 28 ± 10 |

**Fig. 1.** Table 1

[Figure]

**Figure A2.** A) Frequency histograms of the number of nutrient where samples for nitrate concentration were collected at each station and domain: tropical and subtropical Atlantic ocean (red), the Northwestern Mediterranean (green) and Galician coastal upwelling (blue). B) Pair scatter plot representing the relationship between nitrate concentration and density built by using all samples collected during the NICANOR sampling period. C) Frequency histogram of the number of turbulence profiles deployed at each station and domain. D) Pair scatter plot representing the relationship between the euphotic zone depth ($Z_{eu}$) computed using the Morel et al. (2007) equation and the data collected by a radiometer during the HERCULES cruise measured used a radiometer and predicted using the relationship with surface chlorophyll Morel et al. (2007), the solid line represents 1:1 relationship.

**Fig. 2.** Figure A2

[Figure]

**Figure A3.** Vertical distribution of temperature (Temp), nitrate ($NO_3$) and picoplankton biomass of autotrophic (Phyto) and heterotrophic (Bacteria) groups for each domain: tropical and subtropical Atlantic ocean (T), the Northwestern Mediterranean (M), and Galician coastal upwelling (G). Points represent raw data and the solid line the locally weighted scatterplot smoothing (LOESS). Dashed lines indicate 95% confidence intervals. Dot and line color intensity indicates the number of overlapping observations.

**Fig. 3.** Figure A3